# Position: Genomic Model Research Must Move Beyond Anecdotal Evaluation of Interpretability Methods

**Shasha Zhou** [* 1]   **Mingyu Huang** [* 1]   **Ke Li** [1]

## Abstract

Advances in machine learning and computational power have unlocked the predictive potential of the human genome, yet biologists now demand that these models also elucidate the underlying biological mechanisms. While interpretable machine learning (IML) techniques have been increasingly applied to bridge this gap, there has been a pervasive reliance on anecdotal validation: the vast majority of research relies on a single IML method and reports only isolated successful instances. Through a benchmarking study on transcription factor binding, we demonstrate the risks of current practices. We show that different IML methods can often (1) yield contradictory explanations for the same predictions, (2) fail to localize known regulatory motifs, and (3) fail to faithfully reflect the model's internal decision process. In light of this, we argue for a validation framework analogous to clinical trials: just as trials require rigorous design and adverse-event reporting, genomic interpretability must move beyond cherry-picked plausibility toward systematic assessment of consistency, faithfulness, and biological validity. To facilitate this, we propose a tiered framework to guide rigorous evaluation and reporting of genomic IML methods.

## 1. Introduction

Understanding the language of the genome, i.e., how sequence variations shape molecular phenotypes, is a long-standing goal in functional genomics, underpinning applications from prioritizing non-coding variants in human genetics (Avsec et al., 2021a; Zhou et al., 2018), to rational design in synthetic biology (Vaishnav et al., 2022), and

uncovering disease mechanisms for therapeutic target discovery (Gao et al., 2023). To this end, numerous deep learning approaches have proliferated (Zou et al., 2019; Eraslan et al., 2019; Barbadilla-Martínez et al., 2025; Consens et al., 2025; Yang et al., 2025; Hu et al., 2026). With the integration of high-throughput functional assays (Melnikov et al., 2012; Fowler et al., 2010) and burgeoning computational power, these deep neural networks (DNNs) have enabled potent predictive power across various genomic task, such as gene expression (Avsec et al., 2021a; Zhou et al., 2018), transcription factor (TF) binding (Avsec et al., 2021b; Alipanahi et al., 2015; Zhou & Troyanskaya, 2015), enhancer-promoter activity (de Almeida et al., 2022) among many others.

However, the predictive success of these models has not been matched by a commensurate understanding of their inner workings. Because of their intrinsic complexity (e.g., by featuring millions to billions of parameters), such DNNs are often perceived as a "black box", with no explanation about how a given prediction was made or what biological mechanisms were learned. To address this challenge, a variety of *post hoc* interpretability methods have been employed for elucidating the predictions of genomic DNNs (Novakovsky et al., 2023; Chen et al., 2024).

The most prevalent are feature attribution methods such as Saliency Maps (Simonyan et al., 2014), DeepLIFT (Shrikumar et al., 2017), ISM (Zhou & Troyanskaya, 2015), and Integrated Gradients (Sundararajan et al., 2017; Lundberg & Lee, 2017), which quantify the position-specific effect of sequence variations on model predictions. Beyond such main effects, interaction-based methods like integrated Hessians (Janizek et al., 2021), SQUID (Seitz et al., 2024), and DFIM (Greenside et al., 2018) probe the epistatic interactions between mutations, while attention-based interpretation (Vig et al., 2021) and sparse autoencoders (SAEs) (Bricken et al., 2023; Cunningham et al., 2024; Brixi et al., 2026) instead examine the internal computations of transformer-based genomic models. Collectively, these techniques have enabled applications ranging from motif discovery and mechanistic understanding to de novo design and variant prioritization (Avsec et al., 2021b; Alipanahi et al., 2015; de Almeida et al., 2022; Avsec et al.,

---
[*]Equal contribution [1]Department of Computer Science, University of Exeter, Exeter, UK. Correspondence to: Ke Li <k.li@exeter.ac.uk>.

*Proceedings of the 43rd International Conference on Machine Learning*, Seoul, South Korea. PMLR 306, 2026. Copyright 2026 by the author(s).

2021a).

Despite this success, the lack of consensus on how to rigorously evaluate and report interpretability results in genomics is a significant concern. To quantify this gap, we conducted a large-scale mapping study of 3,575 papers employing Interpretable Machine Learning (IML) in genomics between 2010 and 2025. We found that the vast majority of studies: (1) relied on a single IML method without justification; (2) validated interpretations through heterogeneous and often subjective means; and (3) reported only "successful" cases while omitting failures (Section 2).

Furthermore, through a systematic benchmarking study on transcription factor binding prediction, we demonstrate that these practices have tangible consequences. We show that: (a) different IML methods produce drastically inconsistent explanations; (b) these explanations often fail to reflect the model's actual decision-making process; and (c) results frequently misalign with known biological mechanisms—yielding near-zero recovery of true binding sites on challenging tasks, even when cherry-picking "successful" examples remains easy (Section 3).

Because rigorous evaluation is the cornerstone of scientific progress, we argue that genomics researchers must move beyond anecdotal evidence and systematically evaluate IML methods before relying on their outputs.

> **Position:** *The credibility of genomic deep learning is undermined by a reliance on anecdotal evaluation for interpretability. We demonstrate that current practices—characterized by single-method usage and cherry-picked examples—fail to expose that explanations are frequently inconsistent, unfaithful to the model, and biologically misleading. We argue that the field must move beyond subjective "plausibility" toward systematic probing. We propose a tiered framework to rigorously evaluate the* **consistency**, **faithfulness**, *and* **biological validity** *of genomic interpretations.*

## 2. Mapping of Existing Practices

To establish a broad understanding of the existing practices in the evaluation and reporting of IML in genomics, we conducted a comprehensive, data-driven mapping study.

### 2.1. Methods

We first searched on the Web of Science (WoS) database[1] for papers that explicitly mentioned the use of IML methods in genomics by applying the following search query. We applied this query to paper abstracts to optimize the trade-off

---

[1]We use the WoS database because it offers the most comprehensive coverage of literature with highly curated metadata.

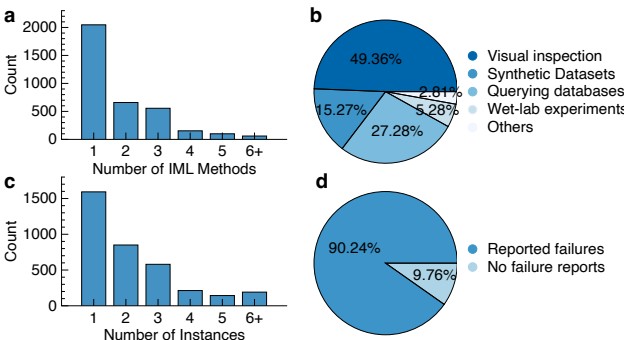

*Figure 1.* **Systematic mapping reveals a reliance on anecdotal evaluation practices.** **(a)** Distribution of the number of IML methods employed per study ($n = 3,575$, the same for all panels). **(b)** Breakdown of validation strategies. **(c)** Frequency of validated interpretation instances per paper. **(d)** Reporting of failure modes.

between coverage and relevance.

```
(
  ("interpretability" OR "explainability"
  OR "explainable AI" OR "XAI"
  OR "feature importance" OR "LIME"
  OR "feature attribution*" OR "SHAP"
  OR "saliency" OR "attention mechanism"
    OR "model explanation*")
 AND
  ("genomics" OR "genome" OR "DNA" OR
    "RNA"
  OR "protein" OR "gene expression"
  OR "sequencing" OR "transcriptomics"
  OR "proteomics")
 AND
  ("machine learning" OR "deep learning"
  OR "neural network*" OR "transformer"
  OR "foundation model*"
  OR "language model*" OR "predict*")
)
```

This search led to 3,575 papers published between 2010 and 2025. Full-text articles were accessed through a combination of institutional subscriptions, open-access repositories, and preprint servers where available. As this amount of papers is too large for manual review, inspired by Huang et al. (2025a), we prompted a `gemini-3-flash-preview` model (to balance costs and accuracy) to extract structured information from each full-text article. Specifically, for each paper, we extracted:

- Types and number of IML methods employed.
  - Any justifications associated with this choice?
- Types of validation for IML performed (e.g., visual inspection, querying domain-specific databases, or via wet-lab experiments?)
- Number of "*successful*" instances, defined as cases the authors *claim* an IML output aligns with established domain knowledge (e.g., an attribution highlighting a known TF binding motif, or identified marker genes

matching known cell types).

– Any report on failed instances and their analysis?

To maximize extraction quality, we employed few-shot in-context learning (ICL) anchored by 10 manually curated demonstrations with self-reflection prompts to avoid hallucinations (see Appendix A for details). We also randomly sampled 100 papers, and one of the authors manually verified the model outputs, which yielded an overall agreement rate of 98% (see Table 2 in Appendix B for the per-category breakdown).

Our analysis reveals several concerning patterns in how IML methods are evaluated and reported in genomics research. First, the vast majority of studies (86%) employed only a single IML method (Fig. 1a), with no comparison against alternative interpretation techniques, nor justifications on why the chosen approach is more appropriate in this case. This lack of cross-method validation makes it difficult to assess the robustness or reliability of the reported interpretations.

Second, and perhaps most concerning, we observed substantial heterogeneity in validation practices (Fig. 1b). Strikingly, nearly half of the studies relied on subjective visual inspection (e.g., whether highlighted regions correspond to known structural contacts). The remaining studies largely relied on indirect proxies like ChIP-seq peaks or simplified synthetic datasets, while only 2.81% conducted wet-lab experiments to validate insights.

Third, the majority of papers reported only 1–3 "*successful*" instances where IML outputs aligned with known biology (Fig. 1c), which raises concerns about cherry-picking. Furthermore, these papers typically did not disclose whether additional instances were examined, nor whether any interpretations failed to align with expectations (Fig. 1d).

Such anecdotal evidence presents a problematic foundation for assessing IML capabilities. The selective reporting of favorable cases risks overstating the reliability of these methods, potentially fostering unwarranted confidence among practitioners. In high-stakes genomics applications—where experimental validation can be prohibitively expensive—over-reliance on inadequately evaluated IML methods could lead to substantial misallocation of resources and, more critically, erroneous biological conclusions.

## 3. Consequences of Current Practices: An Empirical Audit

The mapping study above reveals that evaluation in genomic IML relies heavily on anecdotal evidence. To move beyond abstract critique and concretely demonstrate the risks of these practices, we conduct a targeted empirical audit. We design a controlled case study to illustrate three specific failure modes that arise when rigorous validation is neglected:

*Table 1.* Dataset statistics for each TF. **Ground Truth** denotes the total number of sequences (across splits) with UniBind motif annotations used for interpretability evaluation.

| TF | Train | Validation | Test | Ground Truth |
|---|---|---|---|---|
| CTCF | 95,228 | 4,412 | 3,878 | 42,736 |
| MAX | 110,714 | 4,388 | 4,720 | 31,557 |
| SP1 | 14,764 | 502 | 486 | 1,808 |
| TBP | 41,122 | 1,450 | 1,470 | 2,313 |
| GATA1 | 27,108 | 1,294 | 948 | 10,404 |

(1) **Inconsistency**, where methods yield contradictory explanations; (2) **Unfaithfulness**, where explanations decouple from model logic; and (3) **Biological Misalignment**, where "plausible" explanations fail to match causal ground truth.

**Experimental Design.** We anchor our audit on Transcription Factor (TF) binding prediction (Feng et al., 2025; Avsec et al., 2021b; Vorontsov et al., 2025), a canonical task in regulatory genomics. Rather than aggregating broad metrics, we specifically curated five TFs from ENCODE[2] database to serve as distinct stress tests that probe the boundaries of IML validity:

- **Positive Controls (CTCF, MAX):** We include these factors because they bind long, high-signal motifs. They represent the "easy" cases where any valid explanation method *must* succeed.
- **Compositional Bias Tests (SP1, TBP):** To test whether methods detect specific regulatory grammar or merely "shortcut learn" background frequencies, we selected SP1 (which targets GC-rich islands) and TBP (which targets AT-rich promoters).
- **Resolution Tests (GATA1):** We selected GATA1 to test the spatial precision of token-based architectures, as it binds a short (∼6 bp) and often degenerate motif.

For each transcription factor, we collect high-confidence narrowPeak files from ENCODE. Each peak is represented as a fixed-length DNA sequence centered on the peak summit. Positive samples correspond to experimentally validated binding regions. Negative ones are generated from matched genomic regions that do not overlap called peaks, controlling for chromosome distribution and sequence length.

To ensure these stress tests are statistically robust, we constructed a large-scale benchmark comprising approximately 289,000 training and 11,500 test sequences (summarized in Table 1). We strictly split the data by chromosome, holding out Chromosome 9 entirely for testing to prevent information leakage. Crucially, this scale allows us to move beyond the qualitative inspection of cherry-picked examples; we utilize over 88,000 sequences with annotated motif positions to systematically quantify interpretability performance.

---

[2]https://www.encodeproject.org/

We replicate this audit across three divergent genomic foundation models—**DNABERT-2** (Zhou et al., 2022) (Transformer), **HyenaDNA** (Nguyen et al., 2023) (convolution/SSM), and **Nucleotide Transformer v3** (NTv3) (Boshar et al., 2025) (hybrid)—and evaluate five common interpretability methods: DeepLIFT (Shrikumar et al., 2017), IG (Sundararajan et al., 2017), ISM (Zhou & Troyanskaya, 2015), Local Interpretable Model-agnostic Explanations (LIME) (Ribeiro et al., 2016), and the Categorical Jacobian (CJ) (Zhang et al., 2024). The CJ quantifies pairwise dependencies by measuring how the model's predicted distribution at one position changes under perturbations at every other position; to make it comparable to the four per-position attribution methods, we reduce its position-by-position coupling matrix to a per-position importance score by averaging absolute couplings over partner positions. All three models attain non-trivial predictive performance across the five TFs (see Table 3 in Appendix C for accuracy and F1 scores), ensuring that the explanations we audit are derived from models that have learned meaningful patterns from the data. To establish an objective ground truth, we rely on UniBind[3] database. By retrieving experimentally validated motif coordinates for our held-out test set, we can quantitatively measure whether explanations align with known causal mechanisms, independent of the model's labels.

### 3.1. IML Methods Disagree with Each Other

If IML methods reliably recovered the true explanation for a model's prediction, we would expect different methods to produce consistent results. We test this assumption by computing explanations from all IML methods on the same trained model and comparing their outputs.

**Methodology.** For each test sequence, we obtain per-nucleotide importance scores (absolute values) from each IML method. We quantify agreement between method pairs using: (1) Spearman rank correlation to measure global consistency across all positions, and (2) Jaccard similarity of the top-20 most important positions to assess agreement on critical features—a particularly relevant metric since regulatory motifs are typically shorter than 20 bp.

**Results.** Fig. 2 reveals disagreement among IML methods. Fig. 2a visualizes attribution maps from all five IML methods on the same CTCF sequence on NTv3, showing qualitatively different patterns: DeepLIFT and CJ show a dominant peak with broader background activity across the sequence, IG concentrates almost entirely on a single sharp peak, ISM generates a few sparse and isolated peaks, and LIME produces a diffuse, noisy signal throughout.

Quantitatively, Fig. 2b shows that the average Spearman

---

[3]https://unibind.uio.no/

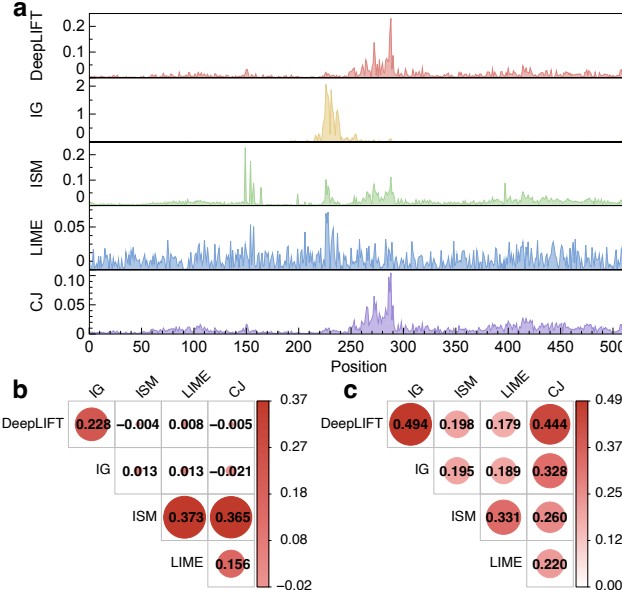

*Figure 2.* **Different IML methods produce inconsistent explanations.** (**a**) Attribution maps from five IML methods on the same CTCF sequence (NTv3 model), showing strikingly different patterns. (**b**) Mean Spearman rank correlation coefficients between method pairs across all models and datasets. (**c**) Mean Jaccard similarity of the top-20 attributed positions between method pairs.

rank correlation between method pairs is consistently below $0.4$ across all models and datasets. The highest correlation (ISM–LIME: $\rho = 0.373$) still indicates weak agreement, while several pairs show near-zero or even negative correlations (IG–CJ: $\rho = -0.021$). Fig. 2c demonstrates that even when focusing on the most critical positions, the maximum Jaccard similarity remains below $0.5$ (DeepLIFT–IG: $0.494$), meaning methods agree on fewer than half of the positions deemed most important. Per-model and per-TF breakdowns of these correlation and Jaccard analyses are provided in Fig. 5 and Fig. 6 in Appendix D.

These results raise a fundamental concern regarding reliability: since different IML methods produce inconsistent explanations, it is unclear which, if any, accurately captures the model's reasoning. If practitioners follow the common practice of using only one IML method, they risk relying on an arbitrary or biased viewpoint that may not reflect the model's true decision process.

### 3.2. IML Explanations Are Not Faithful to Model Decisions

Beyond inter-method consistency, we ask whether explanations faithfully reflect what the *model* considers important—regardless of biological validity. We test this through systematic perturbation experiments (Samek et al., 2016).

**Methodology.** If an IML method correctly identifies posi-

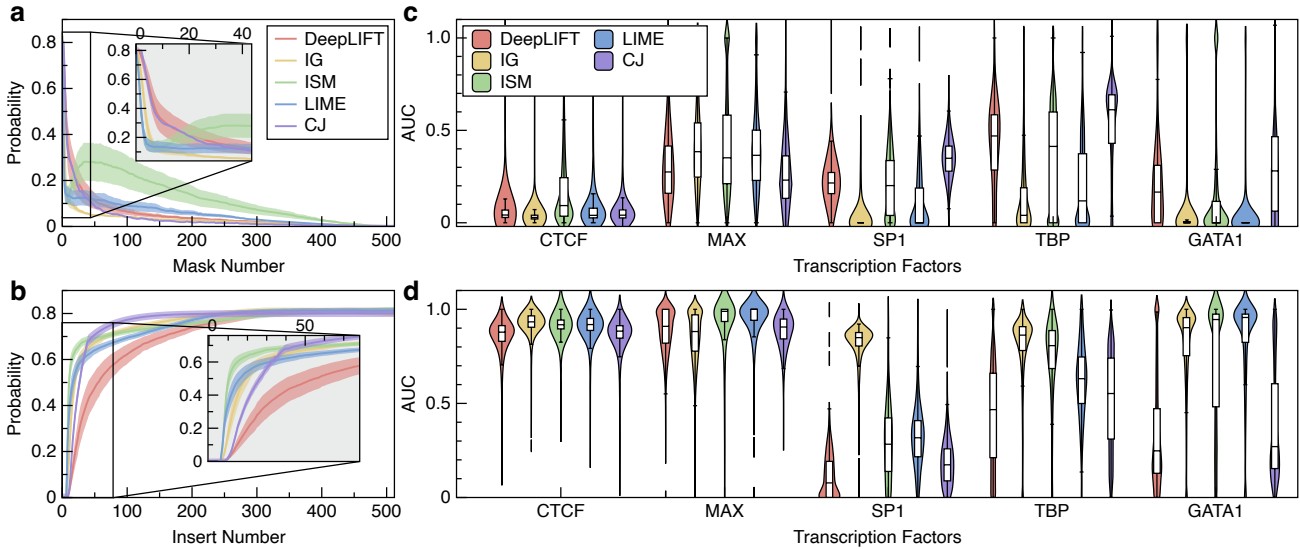

*Figure 3.* **Faithfulness evaluation via perturbation analysis.** (**a**) Sequential deletion (MoRF): prediction probability as top-ranked positions are progressively masked. (**b**) Sequential insertion: probability recovery as positions are restored to a neutral baseline. Curves show means with shaded standard errors on CTCF (NTv3). (**c, d**) Distribution of mean AUC scores across all sequences for deletion (**c**) and insertion (**d**) experiments, stratified by TF on NTv3.

tions most important to a model's prediction, then perturbing those positions should maximally affect model output. We implement two complementary tests:

1. **Sequential deletion (MoRF)** (Samek et al., 2016): Starting from original sequence, we progressively mask positions in descending order of attributed importance by replacing them with neutral tokens('N'). A faithful explanation should produce rapid probability decay.

2. **Sequential insertion** (Petsiuk et al., 2018): Starting from a fully masked baseline, we progressively restore positions in descending order of importance. A faithful explanation should produce rapid probability recovery.

We quantify faithfulness via the area under the probability curve (AUC): lower deletion-AUC and higher insertion-AUC indicate better faithfulness.

**Results.** Fig. 3a and b shows representative curves for CTCF on the NTv3 model. In the deletion experiment, ISM causes the steepest probability drop, followed by LIME and IG, while DeepLIFT and CJ produce the slowest decay. The insets reveal that meaningful separation between methods emerges within the first 20–50 positions—precisely where motif-level features reside. The insertion curves show the inverse pattern: ISM and LIME rapidly recover prediction confidence, while other methods require substantially more positions to achieve comparable recovery.

Fig. 3c and d present the distribution of AUC scores across all sequences and TFs on NTv3 (DNABERT-2 and Hye-

naDNA is shown in Appendix E). For deletion (Fig. 3c), lower AUC indicates better faithfulness; IG consistently achieves the lowest scores across most TFs, though with substantial variance. Notably, for SP1 and TBP—the compositional bias stress tests—all methods show high AUC values clustered near 1.0, suggesting that none reliably identifies the features driving model predictions. The insertion results (Fig. 3d) mirror this pattern: IG generally achieves the highest AUC (indicating faster recovery), but performance degrades markedly on SP1 compared to other TFs.

These results demonstrate that faithfulness varies considerably across both methods and tasks. While IG and ISM show relatively better faithfulness, no method consistently excels, and performance deteriorates on tasks where compositional biases may confound the explanations.

### 3.3. IML Explanations Do Not Align with Biological Ground Truth

Ultimately, the value of IML in genomics lies in recovering biologically meaningful signals. We evaluate whether explanations align with experimentally validated TF motifs.

**Methodology.** For each positive test sequence with an annotated UniBind motif, we extract the contiguous region of length $L$ (matching the motif length) with the highest summed attribution. We then compute the overlap ratio between this predicted region and the ground-truth motif annotation, yielding a Perception score between 0 (no overlap) and 1 (perfect alignment).

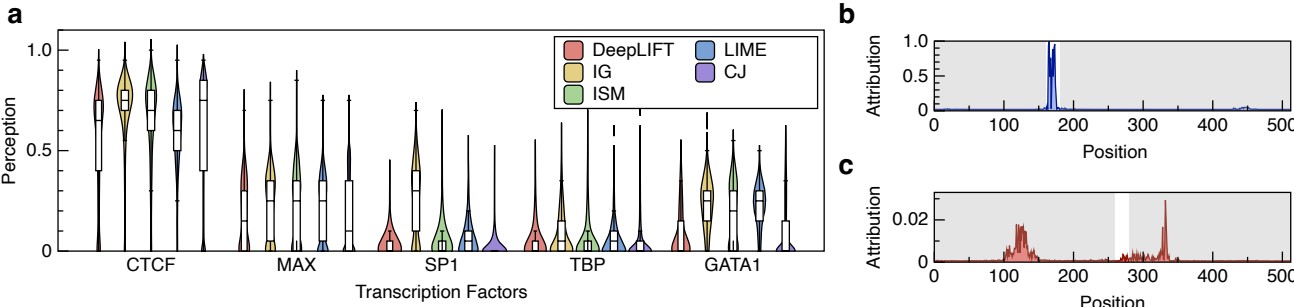

*Figure 4.* **Alignment between IML explanations and biological ground truth.** (**a**) Distribution of motif overlap scores (Perception) across TFs and IML methods on NTv3. Higher values indicate better alignment with UniBind-annotated binding sites. (**b, c**) Representative examples showing attribution profiles (colored curves) relative to ground-truth motif regions (white background): (**b**) successful alignment where attribution peaks coincide with the annotated motif; (**c**) failure case where high attributions occur outside the true binding site.

**Results.** Fig. 4a reveals stark task-dependent variation in biological alignment on NTv3. For CTCF, which is our high-SNR positive control, all methods achieve reasonable alignment, with median Perception scores exceeding 0.5 and many sequences approaching perfect overlap. This confirms that when the biological signal is strong and unambiguous, IML methods can successfully recover it.

However, performance deteriorates dramatically on other TFs. For GATA1 (resolution challenge) and MAX, median scores drop to 0.2–0.4, with substantial probability mass near zero; on GATA1, CJ in particular collapses to a near-zero median, the lowest of all five methods. Most strikingly, for the compositional bias stress tests (SP1 and TBP), all methods show near-complete failure: the vast majority of sequences yield Perception scores close to zero, which indicates that IML-identified important regions rarely coincide with true binding sites. This suggests that explanations are confounded by background nucleotide composition rather than capturing specific regulatory motifs. Per-model Perception distributions are reported in Fig. 10 in Appendix F.

Fig. 4b and c illustrate why anecdotal evidence can be misleading. Fig. 4b shows a good example where LIME attributions (blue) align precisely with the annotated motif region (white background), which is the type of cherry-picked case often showcased in papers. Fig. 4c shows a bad example where high attributions appear entirely outside the true motif location, yet such failures are rarely reported. Our systematic evaluation reveals that failure cases like Fig. 4c are far more common than successes like Fig. 4b for most TF tasks.

## 4. Practical Guidelines for IML Evaluation

With the identified critical limitations in current IML evaluation practices, we now propose a tiered framework of practical guidelines. In particular, considering that practitioners often operate under varying resource constraints, we organize our recommendations by the level of effort required to accommodate different practical scenarios.

### 4.1. Low-Effort Guidelines: Minimum Standards

The following practices require minimal additional effort and should be considered *mandatory* for any study reporting IML results in computational genomics.

**G1: Use Multiple IML Methods.** As demonstrated in the previous section, different IML methods can produce drastically different explanations for the same prediction. Relying on a single method provides no indication of explanation robustness. We recommend:

- Apply at least three IML methods spanning different categories (e.g., gradient-based, perturbation-based, and attention-based) for generating interpretations.
- Report the agreement between methods using quantitative metrics such as rank correlation or top-$k$ overlap, and treat high disagreement as a warning signal that explanations may be unreliable.

**G2: Include Negative Controls.** Explanations should be compared against appropriate baselines to assess whether they provide information beyond chance:

- Compare IML importance scores against random baselines (uniform random importance assignment).
- For sequence data, consider shuffled-sequence controls where the same IML method is applied to sequences with shuffled nucleotide order.

**G3: Report Quantitative Metrics Over Visual Inspection.** Qualitative assessments (*"the highlighted region appears to overlap with the known binding site"*) are subjective and irreproducible. Instead, we recommend:

- Define precise quantitative criteria (e.g., precision, recall, correlation, overlap, etc., when ground-truth annotations are available) *before* examining explanations.

**G4: Report Holistic Statistics Instead of Cherry-Picked Examples.** The common practice of showing 1 to 5 "successful" examples provides no information about typical

explanation quality. Instead, we recommend:

- Compute quantitative evaluation metrics across all test samples.
- Report distributions (mean, standard deviation, percentiles) rather than single exemplars.
- If illustrative examples are shown, explicitly state how they were selected and how representative they are of the overall distribution.

### 4.2. Moderate-Effort Guidelines: Computational Validation

The following practices require additional computational experiments but no wet-lab resources. They provide stronger evidence of explanation quality.

**G5: Perform Faithfulness Tests.** As shown in Section 3, explanations often fail to reflect the model's actual decision process. Faithfulness can be assessed computationally:

- *Perturbation-based evaluation*: Mask or perturb positions identified as important and measure the prediction drop. Compare against random masking.
- *Sufficiency test*: Retain *only* the top-$k$ important positions (masking everything else) and verify that the prediction is preserved.
- *Comprehensiveness test*: Mask the top-$k$ important positions and verify that the prediction degrades substantially.
- Report faithfulness metrics relative to a random baseline. Instead of reporting raw prediction drops, quantify the margin between IML-guided perturbation and random perturbation (e.g., AUC scores).

**G6: Test Across Diverse Conditions.** Explanation quality may vary across different data characteristics. We recommend systematically evaluating across:

- Different *prediction confidence levels* (e.g., do explanations degrade for uncertain predictions?).
- Different *sequence contexts* (e.g., GC-rich vs. AT-rich regions, as in our experiments).
- Different *functional categories* (e.g., promoters vs. enhancers, coding vs. non-coding regions).

### 4.3. High-Effort Guidelines: Experimental Validation

The following practices require wet-lab experiments or substantial additional resources. They provide the strongest evidence but should be guided by computational pre-screening.

**G7: Design Experiments with Proper Controls.** When wet-lab validation is pursued, ensure rigorous experimental design:

- Select validation targets through stratified random sampling across the whole distribution of IML scores, not by cherry-picking high-confidence cases.
- Include both *negative* controls (e.g., instances where the IML method predicts low importance for known functional elements) and *positive* controls (instances where both the model and prior knowledge agree on important regions).
- Pre-register the validation targets *before* conducting experiments to prevent post-hoc selection bias.
- Report all results, including failures and inconclusive cases, not just successful validations.

### 4.4. A Recommended Workflow

We synthesize the above guidelines into a practical workflow for IML evaluation:

1. **Baseline Assessment (Low Effort)**: Apply multiple IML methods, compute agreement metrics, and establish random baselines. If methods strongly disagree or perform near random, *stop*—explanations are likely unreliable.

2. **Computational Validation (Moderate Effort)**: Conduct faithfulness tests and sanity checks. Evaluate against available biological databases. Identify conditions where explanations succeed or fail.

3. **Targeted Experimental Validation (High Effort)**: Based on computational screening, design rigorous wet-lab experiments with proper controls. Validate a representative sample and report all results.

4. **Iterative Refinement**: Use validation results to refine model architecture, training procedure, or IML method selection. Re-evaluate after modifications.

By adopting these tiered guidelines, practitioners can move beyond anecdotal evidence toward rigorous, reproducible evaluation of IML methods. Even implementing only the low-effort guidelines would represent a substantial improvement over current practices and help prevent the resource misallocation documented in Section 2.

## 5. Alternative Views

The concerns we raise about IML evaluation practices may seem overly stringent to some practitioners. Indeed, several alternative viewpoints exist that could justify current practices. We examine each of these perspectives and explain why they are insufficient to ensure reliable interpretability in genomics applications.

**Theoretical Guarantees.** Several popular IML methods come with theoretical foundations. For example, IG satisfies

axioms like sensitivity and implementation invariance (Sundararajan et al., 2017); SHAP values are grounded in cooperative game theory with uniqueness guarantees (Lundberg & Lee, 2017); DeepLIFT provides a principled decomposition of the prediction difference (Shrikumar et al., 2017). However, theoretical guarantees do not translate to practical reliability. The axioms satisfied by these methods concern mathematical properties of the attribution (e.g., that attributions sum to the prediction difference), not whether attributions identify biologically meaningful features (Bilodeau et al., 2022). A method can satisfy all theoretical axioms while still highlighting spurious correlations learned by the model. Moreover, different axiom-satisfying methods can produce drastically different explanations for the same prediction (Krishna et al., 2024; Kindermans et al., 2019; Ghorbani et al., 2019), as we demonstrate empirically in Section 3.1.

**Visual Inspection.** A common practice in the community is to have biologists visually inspect IML outputs on selected examples and assess whether the highlighted regions "make sense" given prior knowledge (Alipanahi et al., 2015; Zhou & Troyanskaya, 2015; Avsec et al., 2021b). This approach suffers from several critical flaws. First, confirmation bias is inevitable: practitioners are more likely to notice and report cases where explanations match expectations, while dismissing or ignoring contradictory cases as noise or edge cases (Kaur et al., 2020; Lakkaraju & Bastani, 2020; Adebayo et al., 2018). Such selective reporting provides no information about the method's reliability in the broader scenarios. Second, visual inspection cannot detect subtle failures—an explanation might highlight a region near but not exactly at the true binding site, which appears correct visually but would fail quantitative evaluation. Finally, this approach cannot assess faithfulness: an explanation might align with biology by coincidence while not reflecting what the model actually learned (Lapuschkin et al., 2019).

**Accuracy Metrics.** If a model achieves high accuracy on test data, one might assume that its explanations must capture true biological signals—otherwise, how could it predict well? This reasoning suggests that evaluation efforts should focus on model accuracy rather than explanation quality. Yet, models can achieve high accuracy through spurious correlations that happen to be predictive in the training distribution but do not reflect causal mechanisms. In genomics, this is particularly concerning: GC content, dinucleotide frequencies, or positional biases can be highly predictive of certain regulatory outcomes without corresponding to specific functional elements (Ghandi et al., 2014; Avsec et al., 2021b; Vorontsov et al., 2025). A model exploiting such shortcuts would produce explanations highlighting these confounders rather than true biological signals (Geirhos et al., 2020; Novakovsky et al., 2023). Furthermore, even when a model does learn true signals, IML methods may fail

to surface them faithfully, as we demonstrate in Section 3.2. Predictive accuracy and explanation quality are orthogonal properties that must be evaluated independently (Adebayo et al., 2018; Rudin, 2019; Sasse et al., 2023).

**IML benchmarks.** The machine learning community has developed various benchmarks for evaluating IML methods in general domains, including synthetic datasets with known ground truth (Hooker et al., 2019) and comprehensive evaluation toolkits (Hedström et al., 2023; Agarwal et al., 2022). While general-purpose benchmarks are valuable, they do not capture the unique challenges of genomics applications (Huang et al., 2025b). Biological sequences have distinct properties—combinatorial motif grammars, long-range dependencies, strand symmetries, and compositional biases—that are absent in natural images or tabular data (Koo & Eddy, 2019; Greenside et al., 2018; Avsec et al., 2021b). IML methods may succeed on standard benchmarks while failing on genomics-specific challenges. For instance, a method might correctly identify important pixels in an image but fail to localize a short motif embedded in a longer regulatory sequence. Domain-specific evaluation using biologically meaningful ground truth is essential to validate IML methods for genomics (Sasse et al., 2023; Feng et al., 2025).

**Wet-lab validation.** While wet-lab validation often provides the strongest evidence, it *alone* cannot serve as the primary evaluation strategy for several reasons. First, experimental validation is expensive and low-throughput—it is infeasible to validate more than a handful of predictions per study. Second, selective validation of high-confidence predictions creates survivorship bias: we only learn about cases where explanations were correct, not the (potentially larger) set of failures. Third, without computational pre-screening, wet-lab resources may be wasted on unreliable explanations. Computational tests serve as essential gatekeepers that identify when explanations are likely unreliable *before* committing experimental resources. The ideal workflow uses computational validation to filter and prioritize candidates for subsequent experimental confirmation.

**Our position.** None of the above alternatives provides adequate assurance that IML explanations are reliable for guiding biological interpretation or experimental design. We therefore argue that practitioners should adopt a multi-layered evaluation strategy as described in Section 4.4. Only through such rigorous evaluation can we move beyond anecdotal evidence and establish genuine confidence in IML-derived biological insights.

## 6. Conclusion

Our results highlight a growing mismatch between how interpretability methods are used in genomic modeling and

how rigorously their outputs are evaluated. Although post hoc IML techniques are often assumed to reveal meaningful model reasoning, our analysis shows that explanations frequently disagree across methods, only weakly reflect model behavior, and often fail to recover known biological mechanisms. These issues are largely obscured by current norms that emphasize qualitative plausibility and selectively reported successes.

This gap matters in practice. Interpretability results increasingly inform experimental prioritization, mechanistic claims, and downstream biological hypotheses. When explanations are unreliable, they risk misdirecting experimental effort and overstating model understanding. We therefore argue that interpretability should be evaluated as an empirical object in its own right. Systematic and rigorous quantitative validation is necessary for interpretability methods to support reliable scientific inference in genomics rather than anecdotal justification.

**Code and Data Availability.** The code and data used in this paper are available at https://github.com/COLA-Laboratory/EvalXAI.

## Acknowledgments

We sincerely thank all the reviewers for their encouraging and constructive feedback. This work was supported by the UKRI Future Leaders Fellowship under Grant MR/S017062/1 and MR/X011135/1; in part by NSFC under Grant 62376056 and 62076056; in part by the Royal Society Faraday Discovery Fellowship (FDF/S2/251014), BBSRC TransformativeResearch Technologies (UKRI1875), Royal Society International Exchanges Award (IES/R3/243136), Kan Tong Po Fellowship (KTP/R1/231017); and the Amazon Research Award and Alan Turing Fellowship. We also acknowledge the compute support provided by Modal.

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

# A. LLM-based Structure Extraction Prompt for the Mapping Study

We use `gemini-3-flash-preview` to extract structured information about interpretable machine learning (IML) evaluation and reporting practices from full-text genomics papers. The whole pipeline can be divided into two stages. At the first stage, we use a few-shot in-context learning approach to extract the structured information from the paper. At the second stage, we use a self-reflection pass to verify the extraction against the actual paper text and correct any errors.

### A.1. Stage 1: Few-shot In-context Learning for Structured Extraction

### A.1.1. SYSTEM INSTRUCTION

```
You are a rigorous scientific information extraction assistant. Your task is to read a
    full-text research paper from the field of genomics / computational biology and
    extract structured information about how interpretable machine learning (IML)
    methods are used, evaluated, and reported in the paper.

You must extract ONLY information that is explicitly stated or directly inferable from
    the paper text. Do NOT guess, speculate, or hallucinate information. If a field
    cannot be determined from the paper, mark it as "unclear".
```

### A.1.2. TASK PROMPT

```
Below is the full text of a research paper from genomics / computational biology.
    Please carefully read the entire paper and extract the following structured
    information about its use of interpretable machine learning (IML) methods.

### Definitions

**Interpretable Machine Learning (IML) methods** include any technique used to explain,
    interpret, or understand the predictions or internal representations of a machine
    learning model. Common examples in genomics include (but are not limited to):
- Feature attribution methods: Saliency Maps, DeepLIFT, Integrated Gradients (IG), In
    Silico Mutagenesis (ISM), DeepSHAP, SHAP, SmoothGrad, GradCAM, Guided
    Backpropagation, LIME
- Interaction methods: Integrated Hessians, DFIM (Deep Feature Interaction Maps), SQUID
- Attention-based interpretation: attention weight visualization, attention rollout
- Concept-based methods: TCAV, concept bottleneck models
- Model-specific interpretation: first-layer filter visualization, motif extraction
    from convolutional filters
- Other post-hoc or intrinsic interpretability approaches

**Validation of IML outputs** refers to any step the authors take to assess whether the
    IML-derived explanations are correct, meaningful, or reliable. This includes:
- **Visual inspection**: Authors or domain experts visually examine IML outputs (e.g.,
    attribution maps, highlighted regions) and qualitatively judge whether they "make
    sense" or align with known biology.
- **Querying databases**: Authors compare IML outputs against entries in biological
    databases (e.g., JASPAR, ENCODE, UniProt, Gene Ontology, KEGG) to check whether
    highlighted features correspond to known biological entities.
- **Synthetic datasets**: Authors use synthetic or simulated data with known
    ground-truth signals to test whether the IML method can recover those signals.
- **Wet-lab experiments**: Authors perform laboratory experiments (e.g., mutagenesis
    assays, CRISPR perturbations, reporter assays, ChIP-qPCR) to validate IML-derived
    predictions.
- **Others**: Any other validation strategy not covered above (e.g., comparison with
    other computational predictions, consistency checks, perturbation-based
    faithfulness tests).

**Successful instance**: A specific case, example, or result that the authors present
    where the IML output aligns with known biology or expected behavior. This is
```

typically shown as a figure or described in text (e.g., "DeepLIFT highlighted the known CTCF binding motif in this sequence").

**Failed instance**: A specific case where the IML output did NOT align with known biology or expected behavior, or where the authors explicitly discuss limitations, errors, or failure modes of the IML method.

### Extraction Fields

For the paper provided, extract the following fields:

1. **iml_methods** (list of strings): List ALL IML methods used in this paper. Use standardized names where possible (e.g., "DeepLIFT", "Integrated Gradients", "ISM", "SHAP", "LIME", "Attention Weights", "GradCAM", "SmoothGrad", "Filter Visualization", etc.). If the paper uses a custom or novel IML method, use the name given by the authors.

2. **iml_method_count** (integer): The total number of distinct IML methods used. Count each method once, even if applied to multiple models or datasets.

3. **iml_justification** (string, one of: "yes", "no", "unclear"): Did the authors provide an explicit justification for why they chose the specific IML method(s)? A justification means a stated reason (e.g., "we used DeepLIFT because it satisfies the completeness axiom" or "we chose SHAP due to its theoretical guarantees"). Simply naming the method without a reason does NOT count as a justification.

4. **validation_types** (list of strings): List ALL types of validation the authors performed on their IML outputs. Use ONLY these categories: "visual_inspection", "querying_databases", "synthetic_datasets", "wet_lab_experiments", "others". If validation involves multiple types, list all that apply. If no validation was performed, use ["none"].

5. **num_successful_instances** (integer 0–5, or "6+", or "unclear"): How many distinct successful instances (specific examples where IML outputs aligned with known biology) are reported in the paper? Count each unique example once. If the count is 6 or more, output "6+". If the authors report aggregate statistics over many instances without showing individual examples, record "0" for individual instances. If unclear, use "unclear".

6. **failure_reported** (string, one of: "yes", "no"): Did the authors report ANY cases where the IML method failed, produced unexpected results, or did not align with known biology? Also counts if the authors discuss limitations or failure modes of the IML outputs (not just general limitations of the ML model).

7. **failure_description** (string or "none"): If failure_reported is "yes", briefly describe what failure was reported. If "no", use "none".

### Output Format

Return your answer as a JSON object with the following structure:

```
{
  "iml_methods": [...],
  "iml_method_count": ...,
  "iml_justification": "yes" | "no" | "unclear",
  "validation_types": [...],
  "num_successful_instances": ...,
  "failure_reported": "yes" | "no",
  "failure_description": "..." | "none"
}
```

A.1.3. FEW-SHOT DEMONSTRATIONS

Ten manually curated demonstrations are provided below. Each consists of an abbreviated paper description and the corresponding gold-standard extraction (annotated by an author).

```
### Few-Shot Demonstrations

Here are 10 manually curated demonstrations. Each consists of an abbreviated paper
    description and the corresponding gold-standard extraction.

#### Demonstration 1

- **Paper description (abbreviated):**
This paper presents a deep learning model for predicting transcription factor binding
    from DNA sequence. The model is a convolutional neural network trained on ChIP-seq
    data. After training, the authors apply DeepLIFT to generate per-nucleotide
    importance scores. They show three example sequences where the DeepLIFT
    attributions highlight the known GATA1 motif (Figures 3A-C). They compare the
    highlighted regions against the JASPAR database and note visual agreement. No cases
    where DeepLIFT failed to recover a motif are discussed. The authors do not explain
    why they chose DeepLIFT over other methods.

- **Gold-standard extraction:**
{
  "iml_methods": ["DeepLIFT"],
  "iml_method_count": 1,
  "iml_justification": "no",
  "validation_types": ["visual_inspection", "querying_databases"],
  "num_successful_instances": 3,
  "failure_reported": "no",
  "failure_description": "none"
}

#### Demonstration 2

- **Paper description (abbreviated):**
This paper develops a transformer-based model for predicting gene expression from
    promoter sequences. The authors use three IML methods: Integrated Gradients,
    attention weight visualization, and in silico mutagenesis (ISM). They justify their
    multi-method approach by stating "we employ multiple interpretability methods to
    ensure robustness of our findings." For validation, they construct synthetic
    sequences with embedded known motifs and test whether each method recovers them.
    They also compare attributions against ENCODE TF binding annotations. They report
    12 examples across three figures where methods agree on highlighting the TATA box
    and SP1 site. They also report that attention weights failed to localize short
    motifs (< 6bp) and discuss this limitation in Section 4.3.

- **Gold-standard extraction:**
{
  "iml_methods": ["Integrated Gradients", "Attention Weights", "ISM"],
  "iml_method_count": 3,
  "iml_justification": "yes",
  "validation_types": ["synthetic_datasets", "querying_databases"],
  "num_successful_instances": "6+",
  "failure_reported": "yes",
  "failure_description": "Attention weights failed to localize short motifs (< 6bp),
      discussed in Section 4.3."
}

#### Demonstration 3

- **Paper description (abbreviated):**
```

```
This paper introduces a graph neural network for protein function prediction. The
    authors use GradCAM to highlight important residues. They show one example (Figure
    5) where GradCAM highlights residues near the active site, and visually inspect it.
    No database comparison is performed. No other IML methods are used. They mention
    using GradCAM because "it is widely used" but do not provide a technical
    justification. No failure cases are reported.

- **Gold-standard extraction:**
{
  "iml_methods": ["GradCAM"],
  "iml_method_count": 1,
  "iml_justification": "no",
  "validation_types": ["visual_inspection"],
  "num_successful_instances": 1,
  "failure_reported": "no",
  "failure_description": "none"
}

#### Demonstration 4

- **Paper description (abbreviated):**
This study applies a recurrent neural network to predict RNA secondary structure. To
    interpret model predictions, the authors use SHAP values and DeepLIFT. They justify
    SHAP as satisfying game-theoretic axioms and DeepLIFT as computationally efficient.
    They compare SHAP attributions against known RNA structural motifs from the Rfam
    database, showing 6 cases of agreement. They also design a synthetic benchmark with
    planted stem-loop structures and evaluate both methods. They find that SHAP
    outperforms DeepLIFT on recovering planted signals. They report two cases where
    DeepLIFT attributions were misleading---highlighting flanking regions instead of
    the true structural motif---and discuss this as a potential issue with
    reference-choice sensitivity.

- **Gold-standard extraction:**
{
  "iml_methods": ["SHAP", "DeepLIFT"],
  "iml_method_count": 2,
  "iml_justification": "yes",
  "validation_types": ["querying_databases", "synthetic_datasets"],
  "num_successful_instances": 6,
  "failure_reported": "yes",
  "failure_description": "DeepLIFT produced misleading attributions in two cases,
      highlighting flanking regions instead of true structural motifs. Authors discuss
      reference-choice sensitivity as the cause."
}

#### Demonstration 5

- **Paper description (abbreviated):**
This paper fine-tunes a genomic foundation model (DNABERT) for variant effect
    prediction. The authors use Integrated Gradients to visualize which nucleotides
    contribute to pathogenicity scores. They show two examples where IG attributions
    overlap with ClinVar-annotated pathogenic variants and visually inspect the
    attribution maps. No explicit justification for choosing IG is provided. They also
    perform a perturbation-based sanity check: masking the top-10 attributed positions
    and observing the prediction drop. They do not report any cases where IG fails. The
    paper mentions using attention weights in supplementary materials to visualize
    layer-wise patterns but does not use them for interpretation of specific
    predictions.

- **Gold-standard extraction:**
{
  "iml_methods": ["Integrated Gradients"],
  "iml_method_count": 1,
```

```
  "iml_justification": "no",
  "validation_types": ["visual_inspection", "querying_databases", "others"],
  "num_successful_instances": 2,
  "failure_reported": "no",
  "failure_description": "none"
}
```

#### Demonstration 6

- **Paper description (abbreviated):**
This paper builds an ensemble model for predicting enhancer activity from DNA sequence.
    No interpretability or explanation methods are applied to the model. The paper
    focuses entirely on prediction performance (AUROC, AUPRC) across multiple cell
    types. The word "interpretability" appears only in the abstract in the context of
    "future work on interpretability."

- **Gold-standard extraction:**
```
{
  "iml_methods": [],
  "iml_method_count": 0,
  "iml_justification": "unclear",
  "validation_types": ["none"],
  "num_successful_instances": 0,
  "failure_reported": "no",
  "failure_description": "none"
}
```

#### Demonstration 7

- **Paper description (abbreviated):**
This paper uses a variational autoencoder for single-cell RNA-seq analysis. For
    interpretability, the authors extract the learned latent space and visualize it
    with UMAP. They also compute SHAP values to identify genes most influential for
    cluster assignments. They validate SHAP outputs by cross-referencing the top-ranked
    genes with known marker genes from the CellMarker database and the Human Protein
    Atlas. They present 5 cell-type-specific gene lists where the top SHAP genes match
    known markers. They also note that for two rare cell types, SHAP highlighted genes
    with no known marker function, which they frame as potentially novel findings
    rather than failures.

- **Gold-standard extraction:**
```
{
  "iml_methods": ["SHAP"],
  "iml_method_count": 1,
  "iml_justification": "no",
  "validation_types": ["querying_databases"],
  "num_successful_instances": 5,
  "failure_reported": "no",
  "failure_description": "none"
}
```

#### Demonstration 8

- **Paper description (abbreviated):**
This paper proposes a new feature attribution method specifically designed for genomic
    sequence models. The authors benchmark their method against DeepLIFT, Integrated
    Gradients, ISM, and LIME on transcription factor binding prediction tasks. They use
    three models (CNN, RNN, Transformer) and five TFs. Validation is performed using
    known motif annotations from JASPAR and UniBind. They report Perception scores
    (motif overlap) across all test sequences for each method and present distribution
    plots. They also perform perturbation-based faithfulness tests (sequential deletion
    and insertion). They show that their method outperforms baselines on 4 of 5 TFs but
    fails on SP1 due to GC-content confounding, which they analyze in detail.

```
- **Gold-standard extraction:**
{
  "iml_methods": ["DeepLIFT", "Integrated Gradients", "ISM", "LIME", "authors' novel
      method"],
  "iml_method_count": 5,
  "iml_justification": "yes",
  "validation_types": ["querying_databases", "others"],
  "num_successful_instances": 0,
  "failure_reported": "yes",
  "failure_description": "The proposed method fails on SP1 due to GC-content
      confounding. Authors analyze this failure mode in detail."
}

#### Demonstration 9

- **Paper description (abbreviated):**
This study trains a multi-task deep learning model to predict histone modifications
    from DNA sequence. The authors apply SmoothGrad and vanilla saliency maps to
    visualize important positions for H3K4me3 predictions. They justify SmoothGrad as
    reducing noise compared to vanilla gradients. They show four example loci where
    attributions overlap with known promoter elements (visual inspection). They also
    query the ENCODE database to check if highlighted regions overlap with DNase-seq
    hypersensitive sites. No failures are mentioned. All four examples are from
    high-confidence predictions.

- **Gold-standard extraction:**
{
  "iml_methods": ["SmoothGrad", "Saliency Maps"],
  "iml_method_count": 2,
  "iml_justification": "yes",
  "validation_types": ["visual_inspection", "querying_databases"],
  "num_successful_instances": 4,
  "failure_reported": "no",
  "failure_description": "none"
}

#### Demonstration 10

- **Paper description (abbreviated):**
This paper applies a pre-trained protein language model to predict protein-protein
    interactions. For interpretability, the authors extract attention maps from the
    final transformer layer and visualize them for 10 protein pairs. They also use
    Integrated Gradients to identify important residues and validate them against known
    interface residues from the PDB (Protein Data Bank). They report that IG correctly
    identifies interface residues for 7 out of 10 protein pairs. For the remaining 3
    pairs, IG highlights allosteric regions distal to the interface, which the authors
    acknowledge as a limitation and attribute to indirect interaction effects. They
    chose IG because "it provides a complete attribution that sums to the prediction
    difference."

- **Gold-standard extraction:**
{
  "iml_methods": ["Attention Weights", "Integrated Gradients"],
  "iml_method_count": 2,
  "iml_justification": "yes",
  "validation_types": ["visual_inspection", "querying_databases"],
  "num_successful_instances": "6+",
  "failure_reported": "yes",
  "failure_description": "IG highlights allosteric regions instead of interface
      residues for 3 out of 10 protein pairs. Authors attribute this to indirect
      interaction effects."
```

```
}
```

### A.1.4. INPUT FORMAT

The full text of the paper is provided below the prompt, enclosed in delimiters:

```
<paper>
{FULL_TEXT_OF_PAPER}
</paper>

Now extract the structured information from this paper following the instructions and
    output format above.
```

## A.2. Stage 2: Self-Reflection Prompt

After Stage 1 returns a JSON extraction for a paper, we will input the paper text and the JSON output back to the LLM to perform the self-reflection. The LLM will re-read the paper and verify each field in the extraction against the actual paper text. It will then correct any hallucinations, miscounts, and misclassifications that slip through Stage 1.

### A.2.1. SYSTEM INSTRUCTION

```
You are a meticulous scientific fact-checker. You will be given a research paper and a
    structured JSON extraction that was previously produced from that paper. Your job
    is to verify every field in the extraction by checking it against the actual paper
    text. You must correct any errors you find. Do NOT trust the previous extraction --
    treat it as a draft that may contain mistakes.
```

### A.2.2. TASK PROMPT

```
Below is a research paper and a JSON extraction that was previously generated from it.
    Your task is to carefully verify each field in the extraction against the paper
    text.

<paper>
{FULL_TEXT_OF_PAPER}
</paper>

<extraction>
{STAGE_1_JSON_OUTPUT}
</extraction>

For EACH of the 7 fields listed below, perform the following verification steps:

---

#### Field 1: iml_methods

- Re-read the paper and list every interpretable machine learning (IML) method actually
    used (not just mentioned in the related work or future directions).
- For each method in the extraction, find the exact passage in the paper where it is
    applied. If you cannot find such a passage, remove it.
- Check for any IML methods used in the paper that are MISSING from the extraction. If
    found, add them.
- Distinguish between: (a) IML methods applied to explain model predictions, versus (b)
    general analysis tools (e.g., UMAP, PCA, clustering) that are not IML methods,
    versus (c) methods only mentioned in passing, related work, or future directions.
- Only count methods in category (a).
```

**Verification output**: List the corrected iml_methods, and for each method, quote or
    paraphrase the passage where it is used.

---

#### Field 2: iml_method_count

- Recount based on your verified iml_methods list.
- Ensure each method is counted exactly once, even if applied to multiple models,
    datasets, or tasks.

**Verification output**: Corrected count.

---

#### Field 3: iml_justification

- Search the paper for any explicit reason the authors give for choosing their IML
    method(s).
- A valid justification includes: theoretical properties (e.g., "satisfies completeness
    axiom"), empirical properties (e.g., "shown to be more faithful in prior work"),
    practical considerations (e.g., "computationally efficient for our architecture"),
    or systematic comparison goals (e.g., "we benchmark multiple methods to assess
    robustness").
- Statements like "we use X" or "X is a popular method" without a reason do NOT count.
- If the paper uses multiple methods as a deliberate comparison/benchmark, that framing
    itself counts as a justification.

**Verification output**: "yes", "no", or "unclear", with the supporting quote or
    explanation.

---

#### Field 4: validation_types

- For each validation type in the extraction, verify that the paper actually performs
    that type of validation on the IML outputs (not on the ML model's predictions).
- Key distinction: validating that a model predicts well (e.g., AUROC on test set) is
    NOT validation of IML outputs. Validation of IML outputs means checking whether the
    explanations are correct, faithful, or biologically meaningful.
- Check for validation types that are present in the paper but MISSING from the
    extraction.
- Use only these categories: "visual_inspection", "querying_databases",
    "synthetic_datasets", "wet_lab_experiments", "others", "none".

**Verification output**: Corrected list, with a brief note for each type explaining
    what the paper actually does.

---

#### Field 5: num_successful_instances

- Count the number of distinct, individually presented examples where the authors show
    that an IML output aligns with known biology or expected behavior.
- Count figures, subfigures, or text descriptions that each present a separate case.
- Do NOT count aggregate statistics (e.g., "average overlap across 1000 sequences") as
    individual instances -- those count as 0 individual instances.
- If the count is 6 or more, output "6+".
- If ambiguous, output "unclear".

**Verification output**: Corrected count, with a brief justification of how you counted.

---

```
---

#### Field 6: failure_reported

- Search the paper for any discussion of cases where the IML method failed, produced
    unexpected results, or did not align with known biology.
- Also counts: explicit discussion of limitations specific to the IML outputs (not
    general model limitations).
- Does NOT count: authors framing unexpected results as "novel findings" or
    "interesting observations" without acknowledging them as potential failures.

**Verification output**: "yes" or "no", with the supporting passage or explanation.

---

#### Field 7: failure_description

- If failure_reported is "yes", provide a concise summary of the failure.
- If "no", output "none".
- Verify that the description accurately reflects what the paper says -- do not
    exaggerate or understate.

**Verification output**: Corrected description or "none".

---

### Final Output

After verifying all 7 fields, produce your output in the following JSON format:

{
  "verified_extraction": {
    "iml_methods": [...],
    "iml_method_count": ...,
    "iml_justification": "yes" | "no" | "unclear",
    "validation_types": [...],
    "num_successful_instances": ...,
    "failure_reported": "yes" | "no",
    "failure_description": "..." | "none"
  },
  "changes_made": [
    "Describe each change you made, e.g.: 'Removed X from iml_methods because it was
        only mentioned in related work, not applied.'"
  ],
  "confidence": "high" | "medium" | "low"
}

Rules:
- If you find NO errors, return the original extraction unchanged under
    "verified_extraction" and set "changes_made" to ["No changes needed."].
- If you are unsure about a field, err on the side of "unclear" rather than guessing.
- The "confidence" field reflects your overall confidence in the final verified
    extraction: "high" means all fields have clear textual support; "medium" means 1-2
    fields required judgment calls; "low" means significant ambiguity remains.
```

## B. Per-Category Agreement Rates of LLM-based Extraction

To assess the reliability of the LLM-based extraction pipeline described in Appendix A, one of the authors manually annotated a random sample of 100 papers and compared the manual labels against the LLM outputs across the four extraction categories used in our mapping study. The per-category agreement rates are reported in Table 2. The overall agreement rate, averaged across all categories, is 98%.

*Table 2.* Agreement rates between manual annotation and LLM extraction across evaluated categories.

| Extraction Category | Agreement Rate (%) |
|---|---|
| Number of IML methods | 100 |
| Validation strategy type | 96 |
| Number of reported instances | 97 |
| Failure reporting (yes/no) | 99 |

## C. Predictive Performance of Foundation Models on TF Binding Prediction

Table 3 reports the test-set predictive performance (accuracy and F1 score) of the three genomic foundation models—DNABERT-2, HyenaDNA, and NTv3—fine-tuned on the TF binding prediction task across the five transcription factors (CTCF, MAX, SP1, TBP, and GATA1) introduced in Section 3. These results establish that all three models attain non-trivial predictive performance, which is a prerequisite for the subsequent interpretability audit: the explanations evaluated in our study are derived from models that have learned meaningful patterns from the data.

*Table 3.* Predictive performance (Accuracy and F1 score) of the three foundation models on TF binding prediction across five transcription factors.

| Model | CTCF | | MAX | | SP1 | | TBP | | GATA1 | |
|---|---|---|---|---|---|---|---|---|---|---|
| | Acc | F1 | Acc | F1 | Acc | F1 | Acc | F1 | Acc | F1 |
| DNABERT-2 | 0.804 | 0.823 | 0.587 | 0.684 | 0.774 | 0.772 | 0.797 | 0.770 | 0.832 | 0.821 |
| HyenaDNA | 0.916 | 0.916 | 0.876 | 0.875 | 0.790 | 0.794 | 0.797 | 0.780 | 0.915 | 0.916 |
| NTv3 | 0.950 | 0.951 | 0.885 | 0.888 | 0.872 | 0.876 | 0.853 | 0.849 | 0.941 | 0.942 |

## D. Per-Model and Per-TF Breakdown of IML Disagreement

Fig. 5 and Fig. 6 report the inter-method Spearman rank correlation and top-20 Jaccard similarity, broken down by foundation model (DNABERT-2, HyenaDNA, NTv3) and by transcription factor (CTCF, MAX, SP1, TBP, GATA1). These breakdowns complement the model- and TF-aggregated heatmaps in Fig. 2b and c, and show that the low pairwise agreement reported in 3.1 is not driven by a particular model or TF.

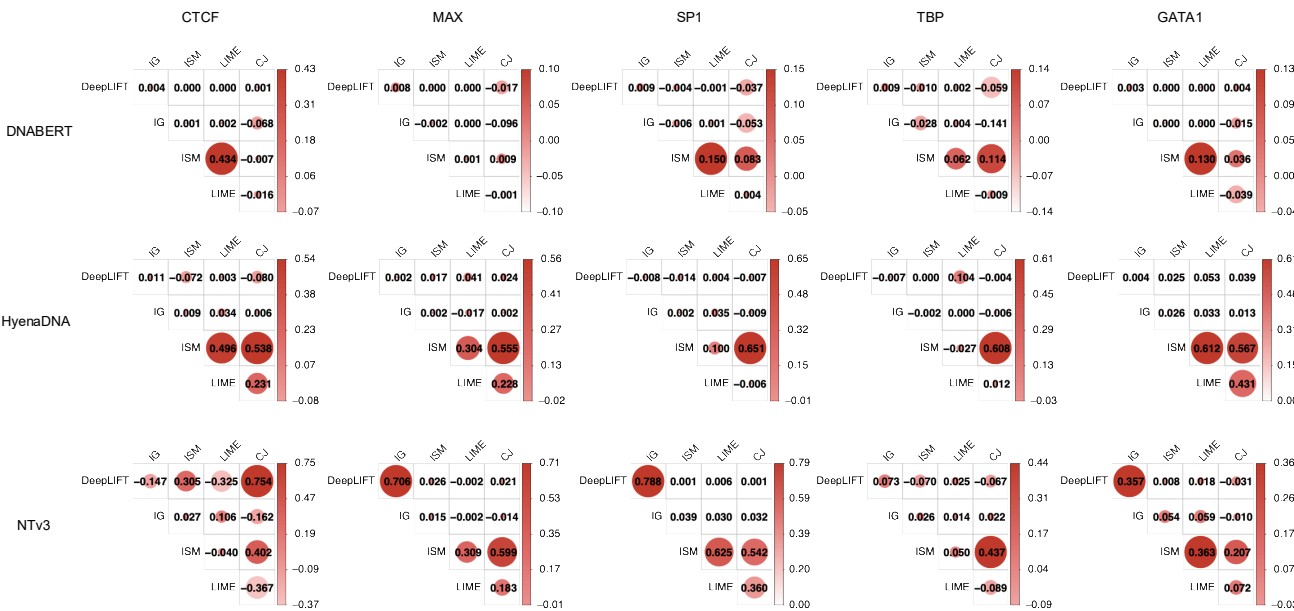

*Figure 5.* **Per-model and per-TF Spearman rank correlation between IML method pairs.** Each subplot reports the pairwise Spearman correlation matrix over the five interpretability methods (DeepLIFT, IG, ISM, LIME, CJ) for one combination of foundation model (DNABERT-2, HyenaDNA, NTv3) and transcription factor (CTCF, MAX, SP1, TBP, GATA1).

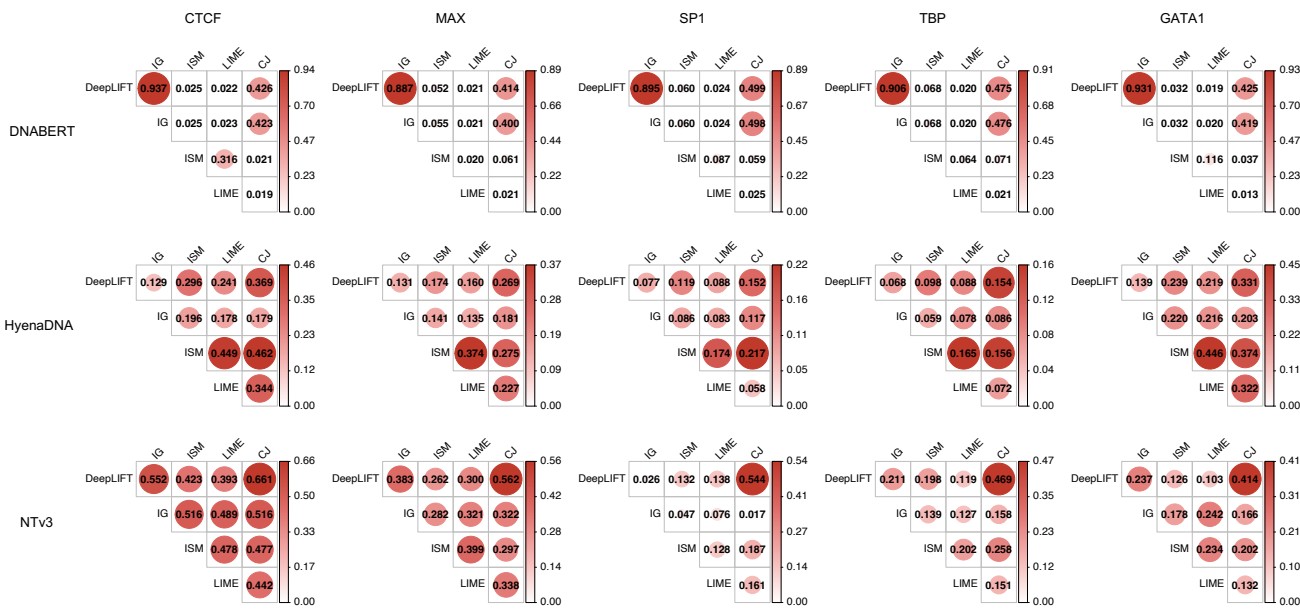

*Figure 6.* **Per-model and per-TF top-**20 **Jaccard similarity between IML method pairs.** Each subplot reports the pairwise Jaccard similarity of the top-20 attributed positions over the five interpretability methods for one combination of foundation model (DNABERT-2, HyenaDNA, NTv3) and transcription factor (CTCF, MAX, SP1, TBP, GATA1).

## E. Per-Model Faithfulness Evaluation

Fig. 7, Fig. 8, and Fig. 9 report the deletion-AUC and insertion-AUC distributions on DNABERT-2, HyenaDNA, and NTv3, respectively.

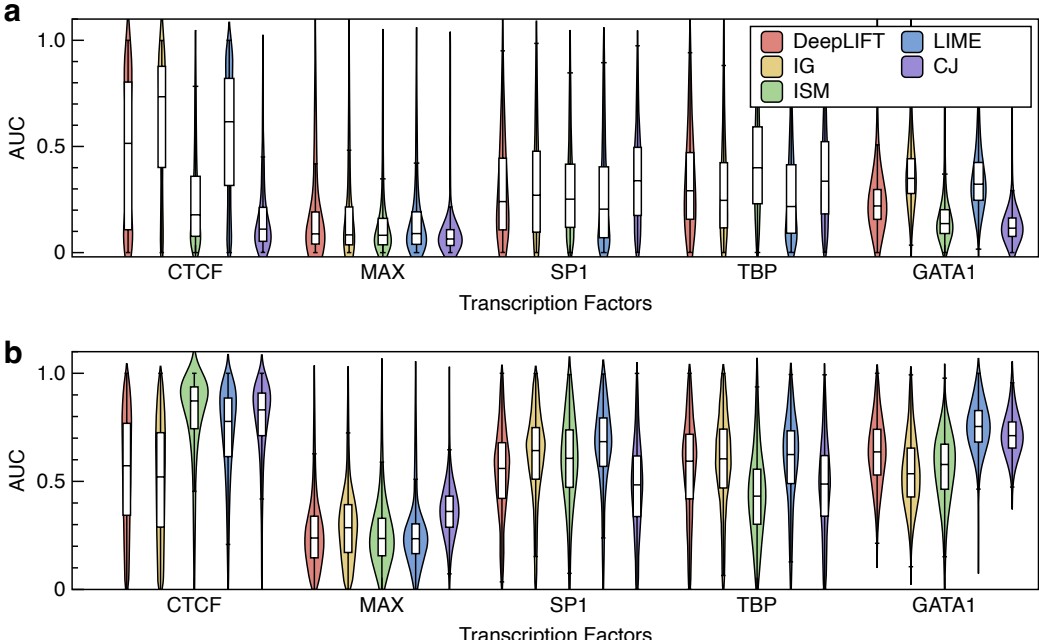

*Figure 7.* **Faithfulness AUC distributions on DNABERT-2.** Distribution of mean AUC scores across all test sequences, stratified by transcription factor (CTCF, MAX, SP1, TBP, GATA1) and IML method, for (**a**) the deletion experiment (lower AUC indicates better faithfulness) and (**b**) the insertion experiment (higher AUC indicates better faithfulness).

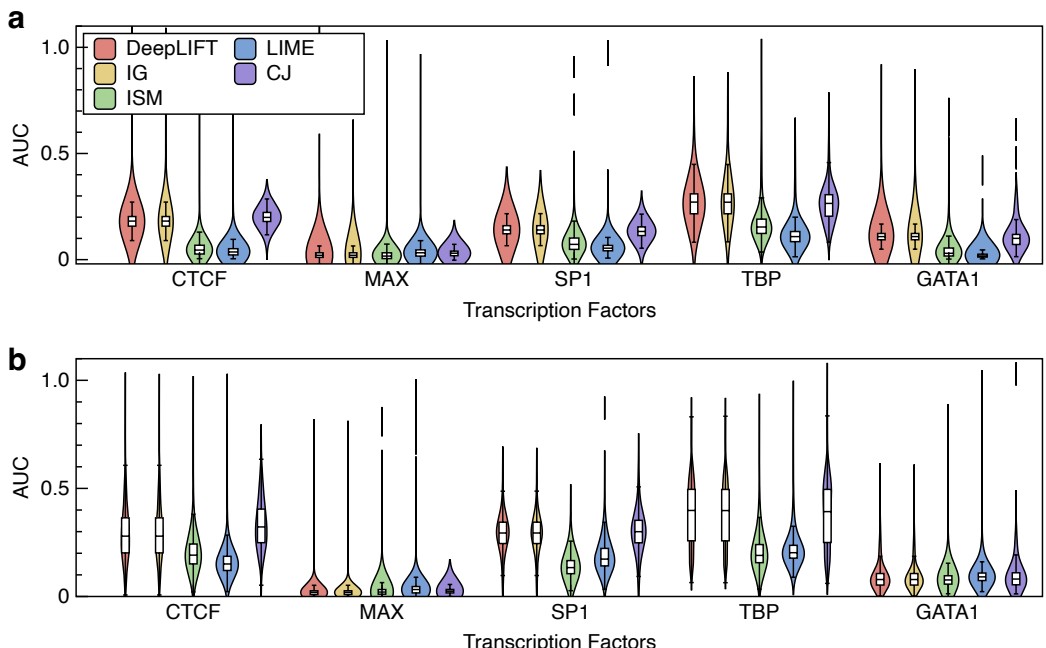

*Figure 8.* **Faithfulness AUC distributions on HyenaDNA.** Distribution of mean AUC scores across all test sequences, stratified by transcription factor (CTCF, MAX, SP1, TBP, GATA1) and IML method, for (**a**) the deletion experiment (lower AUC indicates better faithfulness) and (**b**) the insertion experiment (higher AUC indicates better faithfulness).

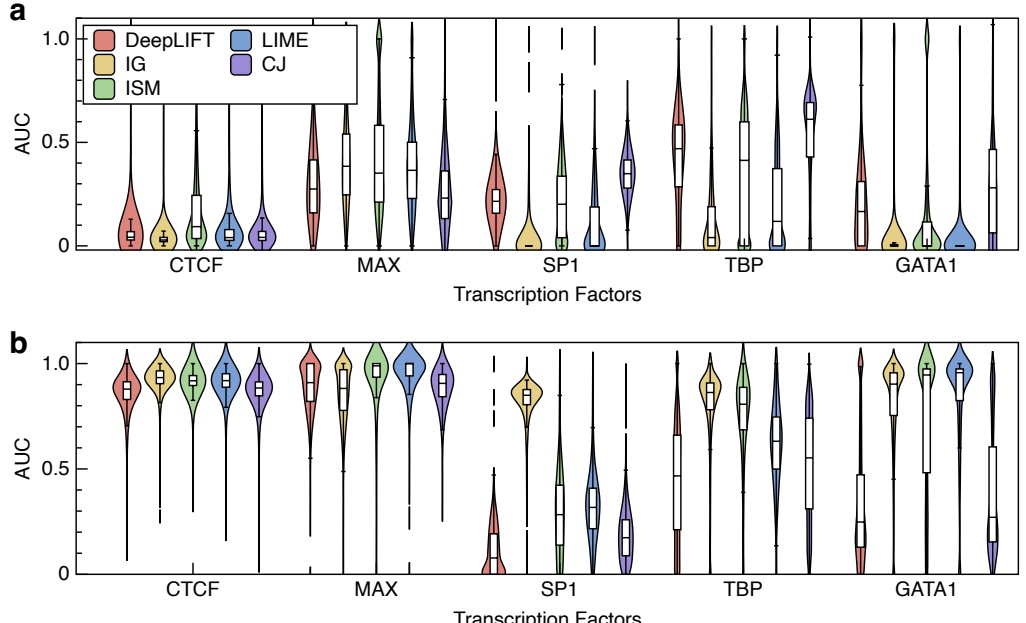

*Figure 9.* **Faithfulness AUC distributions on NTv3.** Distribution of mean AUC scores across all test sequences, stratified by transcription factor (CTCF, MAX, SP1, TBP, GATA1) and IML method, for (**a**) the deletion experiment (lower AUC indicates better faithfulness) and (**b**) the insertion experiment (higher AUC indicates better faithfulness).

## F. Per-Model Biological Alignment

Fig. 10 reports the per-TF and per-IML method Perception scores on DNABERT-2, HyenaDNA, and NTv3.

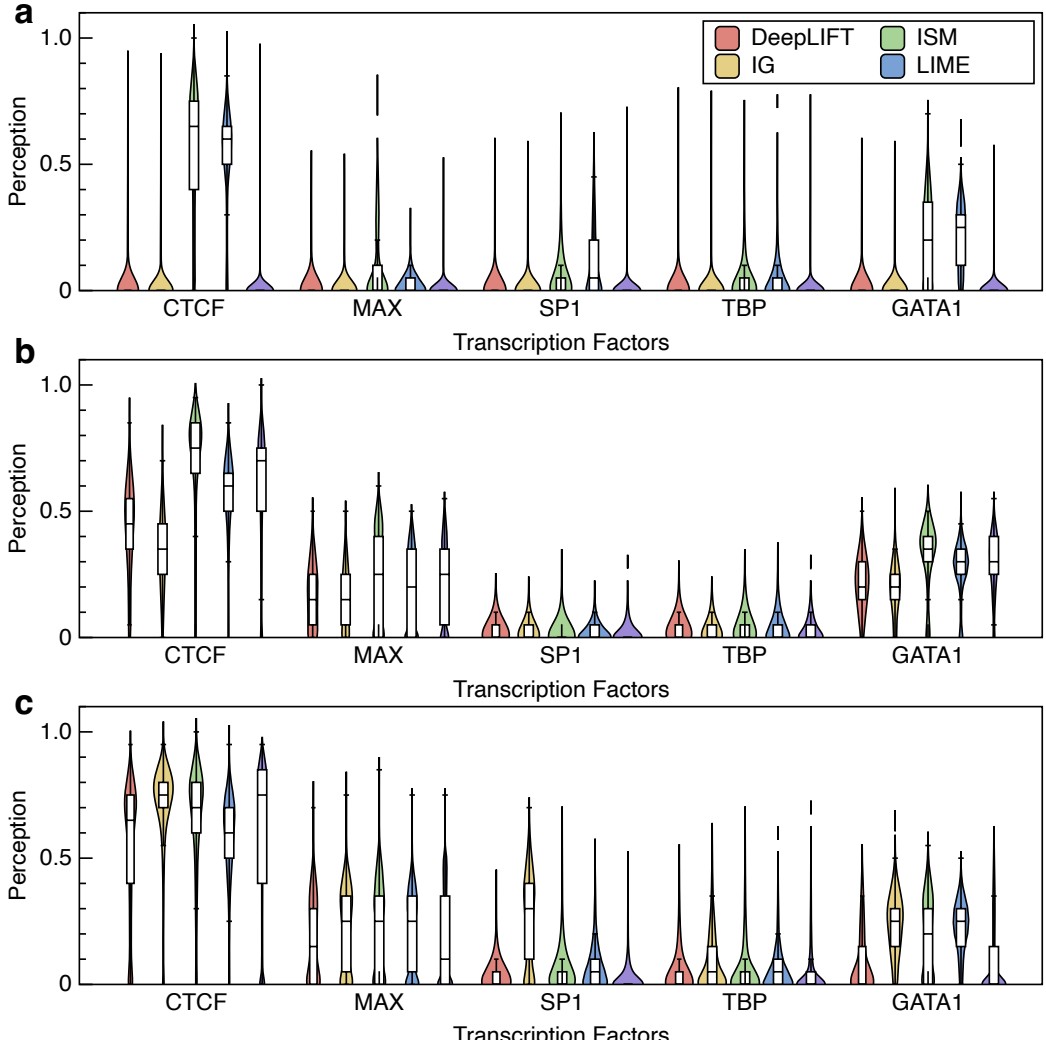

*Figure 10.* **Per-model alignment between IML explanations and biological ground truth.** Distribution of motif overlap (Perception) scores across the five transcription factors (CTCF, MAX, SP1, TBP, GATA1) and the five IML methods (DeepLIFT, IG, ISM, LIME, CJ), shown separately for (**a**) DNABERT-2, (**b**) HyenaDNA, and (**c**) NTv3. Higher values indicate better alignment with UniBind-annotated binding sites.

