# OpenReview forum: "Position: Genomic Model Research Must Move Beyond Anecdotal Evaluation of Interpretability Methods"
_ICML.cc/2026/Position_Paper_Track — ICML 2026 Position Paper Track spotlight_

### Official Review · Reviewer_HW7Y · 2026-02-22

**Significance:** 4
**Argument Clarity:** 4
**Rating:** 5
**Confidence:** 4

**Questions:**

I would like you to state what sort of claims you think interpretability methods can actually be used to support. Especially claims which cannot be supported by other more straight-forward means.

**Alternative Views Section:**

Yes

**Compliance With Llm Reviewing Policy A Conservative:**

Affirmed.

**Discussion Potential:**

4

**Final Justification:**

My prior assessment of strengths and weaknesses remains accurate. I keep my score and confidence, and still recommend accept.

**Paper Summary:**

The paper highlights issues in the ways that interpretability methods are used in genomic deep learning papers. First they canvas a large set of genomic deep learning papers and show that papers that use interpretability methods do not regularly compare between interpretability methods, do not show many examples, do not bother to validate whether the output of the model is correct beyond visual inspection or a database lookup.

Next they argue that these practices can lead to poor conclusions through a carefully build synthetic example. They also later explicitly reconcile these findings with high benchmark scores and theoretical guarantees that interpretability methods bring.
Finally they suggest a set of guidelines that would allow practitioners that their analysis does not suffer from these pathologies.

**Position:**

Yes

**Position In Title:**

Yes

**Related Work:**

4

**Strengths And Weaknesses:**

Strengths:

* The argument is very clear.

* The alternative views section is additive to the paper.

* The evidence presented is an effortful mix of literature search and synthetic study. The results of both were surprising to me.

Weaknesses:

Perhaps the main weakness I'd like to bring up is that the use of interpretability methods is not really brought up in context. With context, I think the guidelines can sometimes seem inappropriate.

For instance, imagine I want to suggest that my model makes use of the presence of a particular motif to make a prediction. I need to get the motif from somewhere, so perhaps I use an interpretability method to get a motif that I proceed to do in silico experiments on. In this setting, I perform the "medium" experiments in 4.2, but for my claim it would not help me to perform the "easy" experiments.

Interpretability methods, used for any application other than hypothesis generation, seems to me, maybe naively, to be inappropriate. Therefore, one should never bother doing the experiments in 4.1, and whether to do 4.2 or 4.3 would have to do with the hypothesis.

Ultimately, a discussion about how interpretability methods can be used in the logic of the claims of a paper (if at all), and connecting that with section 4, would strengthen the paper for me.

Typo: line 126 should be Fig 1b

**Support:**

3

---

> ### Author Rebuttal · Authors · 2026-03-27
>
> ***Abbreviations:** **"W"** for Weaknesses, **"Q"** for Questions.*
>
> We thank the reviewer for recognizing our manuscript and for their constructive feedback. Below, we address each comment in detail.
>
> ---
> ### **W1: Necessity of Low-Effort Guidelines**
>
> We thank the reviewer for this insightful point. We agree that downstream validation ("medium" in 4.2 and "high" in 4.3) is the ultimate test for an IML hypothesis, especially in the mentioned contexts.
>
> However, these validations are often expensive, requiring either large-scale in-silico (4.2) or wet-lab experiments (4.3). Therefore, **it is infeasible to experimentally validate all IML hypotheses**, which makes the low-effort guidelines in 4.1 essential – **they provide a cost-effective filter to detect unreliable IML outputs before committing to expensive validation.**
>
> For example, as shown in Table 1 below, almost all IML methods fail to recover valid motifs for SP1 and TBP. A practitioner who skipped the low-effort agreement (4.1) and faithfulness (4.2) checks would have no warning that their hypothesis was spurious. However, running these checks reveals remarkably low agreement and faithfulness scores, thus immediately flagging these outputs as high-risk and preventing waste of expensive downstream experimental resources.
>
> **Table 1:** Model performance and inter-method agreement, faithfulness (insertion-AUC), as well as biological alignment averaged across 4 different IML methods on TF binding prediction. Other setups are the same as in Section 3.
> TF|NTv3 Performance (F1↗)|IML Agreement(↗)|IML Faithfulness(↗)|IML Biological Alignment(↗)
> -|-|-|-|-
> CTCF|0.95|0.49(±0.19)|0.89(±0.07)|0.62(±0.21)
> MAX|0.88|0.35(±0.16)|0.87(±0.10)|0.28(±0.16)
> SP1|0.87|**0.15(±0.18)**|**0.26(±0.19)**|**0.09(±0.14)**
> TBP|0.85|**0.20(±0.18)**|**0.59(±0.25)**|**0.17(±0.13)**
> GATA1|0.94|0.20(±0.19)|0.42(±0.31)|0.27(±0.11)
>
> **`Action 1:`** We will revise Section 4 to clarify the relationship between different tiers of guidelines.
>
> ---
> ### **W2/Q1: Contextualizing IML Claims and Our Guidelines**
>
> **1. Supported Claims in IML**
>
> We thank the reviewer for this constructive suggestion. We agree that this discussion would strengthen the manuscript. To this end, during the rebuttal, we employed the LLM-assisted literature mapping pipeline (Section 2) to inspect the role of IML across our 3,575 collected papers. Our findings confirm the reviewer's point that "hypothesis generation" broadly summarizes IML usage. Building on this, we categorize these applications into two major types of tasks:
>
> - **Downstream Tasks (Biological Side):** This represents the most standard usage of IML, which uses IML outputs as potential novel hypotheses for diverse biological discovery scenarios.
>
> - **Upstream Tasks (ML Side):** IML can also provide insights in ML development workflows to guide iterative refinements, including:
>   - Model sanity checking, e.g., *"does the model rely on genuine biological signals rather than dataset artifacts?"*, addressing "shortcut learning" in lines 405-425;
>   - Model debugging, e.g., *"does this feature drive the model's false predictions?"*;
>   - Model comparison & selection, e.g., *"does our proposed architecture outperform the baseline because it successfully captures long-range epistatic interactions?"*
>
> **2. Why Other Straightforward Means are Insufficient**
>
> IML facilitates both application scenarios that other straightforward means would not suffice:
> - **For upstream tasks:** Standard model performance metrics (e.g., accuracy, loss) merely quantify overall predictive power without revealing the model's internal logic behind each prediction. This opacity prevents granular sanity checks and informed debugging as discussed above.
> - **For downstream tasks:** IML bridges the historical divide between interpretable statistical methods (or classic ML models) and high-performing, black-box DNNs by distilling complex learned patterns into potential biological hypotheses.
>
>
> **3. Connecting Claims to the Tiered Guidelines**
>
> Building on W1, basic checks (Section 4.1) are necessary for both IML applications, while we agree with the reviewer's point that the focus on 4.2 or 4.3 differs by task:
> - **For upstream tasks**, faithfulness checks in 4.2 are essential to ensure that the evidence from IML truly reflects the model's reasoning. For some studies that demand stronger evidence, further validation with biological experiments (4.3) may also be necessary. In such cases, as shown in Table 1, 4.1 and 4.2 should act as gatekeepers to minimize wasted experimental resources.
> - **For downstream tasks**, experimental validation in 4.3 is the gold standard. Still, 4.1 and 4.2 remain necessary to filter out unreliable hypotheses before expensive experiments.
>
> **`Action 2:`** We will revise Section 2 to include the results on common IML claims and revise Section 4 to contextualize the guidelines with these.
>
> ---
> ### **W3: Typo**
>
> Thank you for catching the typo; we have revised it.

---

> > ### Author Rebuttal · Reviewer_HW7Y · 2026-04-03
> >
> > I really appreciate the new analysis in action 2. Your paper is fine, I think the analysis you've done has a lot to add to the community. But I think a deeper connection between uses and guidelines is necessary.
> >
> > For downstream analysis for example, you suggest "4.1 and 4.2 remain necessary to filter out unreliable hypotheses before expensive experiments". Why? It seems strange to put guidelines on hypotheses directly. Consider that these models may simply give a scientist an idea for a biophysically plausible mechanism, in which case the scientist's domain expertise is the filter.
> >
> > For "upstream" tasks, none of 4.1, 2, or 3 can guarantee the behaviour of the model, or that of the features it "uses". IML is the wrong tool for reliability in this way; instead methods that build explicit frequentist guarantees, such as those int he family of conformal inference, are the appropriate tool.
> >
> > I think there's an unmentioned desire in your writing to use the outputs of IML as *evidence* for a hypothesis. For your paper to argue something like this, even though you're taking a sceptical stance, I think you should show how if IML satisfies 4.1 then it can serve as statistical evidence.

---

### Official Review · Reviewer_vbqK · 2026-03-12

**Significance:** 3
**Argument Clarity:** 3
**Rating:** 4
**Confidence:** 4

**Questions:**

- Do you believe your stratified guidelines should be adjusted for different types of genomics tasks, or do you advocate uniformly high standards across all scenarios?

**Alternative Views Section:**

Yes

**Compliance With Llm Reviewing Policy A Conservative:**

Affirmed.

**Discussion Potential:**

3

**Final Justification:**

The rebuttal addressed my main concerns.

**Paper Summary:**

The core position of the paper is: Current practices relying on single methods, subjective validation, and selective reporting severely undermine the credibility of deep learning models in genomics, and the field must shift toward systematic quantitative evaluation. Through a large-scale mapping study of 3,575 papers published between 2010 and 2025, the authors uncover three major flaws in current practice. The authors argue that genomic interpretability research should move beyond anecdotal evidence and adopt rigorous validation frameworks analogous to clinical trials. They propose a stratified practical guideline for systematically evaluating consistency, faithfulness, and biological validity.

**Position:**

Yes

**Position In Title:**

Yes

**Related Work:**

2

**Strengths And Weaknesses:**

Strengths:

- This paper addresses a critical bottleneck in deep learning for genomics: predictive performance far outpaces mechanistic understanding. With the widespread adoption of foundation models such as DNABERT-2 and Nucleotide Transformer in genomics, the reliability of IML methods directly impacts biological hypothesis generation and experimental resource allocation.

- The paper employs a triangulation strategy: a large-scale bibliometric analysis (3,575 papers) establishes the pervasiveness of the problem; controlled benchmarks (5 transcription factors, 3 foundation models, 4 IML methods) demonstrate concrete failure modes.

- The paper systematically examines five potential alternative viewpoints (theoretical guarantees, visual inspection, accuracy metrics, generic IML benchmarks, and wet-lab validation) and explains why none can substitute for systematic evaluation. This preemptive argumentation reinforces the robustness of the position.


Weaknesses:

- While the Gemini-3-flash-preview model is used to extract structured information from 3,575 papers, some details are missing. For example, the exact design of the prompting strategy. Given potential biases in LLM-based extraction, these details are critical for assessing the reliability of the mapping study.

- The paper criticizes current practice for only reporting “successful” cases, yet does not clearly define what constitutes success.

**Support:**

3

---

> ### Author Rebuttal · Authors · 2026-03-27
>
> ***Abbreviations:** **"W"** for Weaknesses, **"Q"** for Questions.*
>
> We thank the reviewer for the constructive feedback. Below, we address each comment in detail.
>
> ---
> ### **W1: Details on LLM-Based Mapping Study**
>
> We thank the reviewer for this constructive comment, and we address the concerns from the following aspects:
>
> - **Raw Prompts:**  We have made the complete extraction prompts available at [this link](https://doi.org/10.5281/zenodo.19340974).
>
> - **Strategies to Mitigate LLM Bias:** We agree that LLMs can be susceptible to extraction bias. To mitigate this with a reasonable budget, we utilized two primary strategies (lines 110-113): first, we included 10 expert-crafted demonstrations to align the model with our extraction goals; second, we employed a self-reflection step, requiring the LLM to critically analyze its output for hallucinations or omissions before finalizing the response.
>
> - **Manual Validations:** To verify LLM adherence (Section 2.1, lines 113-115), an author independently annotated 100 random papers. As detailed in Table 1 below, we observed high overall alignment between the manual and LLM extractions. In addition, we found that most discrepancies stemmed from ambiguous writing in the source paper, not LLM hallucinations or bias.
>
> **Table 1:** Agreement rates between manual annotation and LLM extraction across evaluated categories.
> | Extraction Category | Agreement Rate (%) |
> |---|---|
> | Number of IML methods |100|
> | Validation strategy type | 96|
> | Number of reported instances | 97 |
> | Failure reporting (yes/no) | 99|
>
> **`Action 1:`** We will include the extraction prompt in the Appendix to enhance reproducibility.
>
> ---
> ### **W2: Definition of "Successful Cases" in Mapping Study**
>
> We thank the reviewer for this comment. In our mapping study in Section 2, **a "successful explanation" is one where the IML's output is *claimed* to align with domain knowledge.** For instance, a success occurs when a feature attribution method highlights a DNA sequence region that overlaps with a known functional binding motif, or when identified marker genes correspond perfectly to known cell types.
>
> We further note that as shown in Fig. 1b, **literature frequently verifies *claimed* successes subjectively** (e.g., by visual inspection), which hampers reproducibility and cross-study comparison. In contrast, our empirical experiments in Section 3 utilize quantitative assessments against biological ground truths, enabling a simultaneous comparison of multiple IML methods across various models and tasks. This underscores Section 4's guidelines on quantitative metrics for IML evaluation.
>
> **`Action 2:`** We will revise our manuscript to explicitly describe the concept of "successful cases" in Section 2.
>
> ---
> ### **Q1: Should Guidelines Be Task-Specific?**
>
> The reviewer has raised a great question. We do not intend to advocate a one-size-fits-all validation burden across all genomics tasks. Rather, our view is that **the low-effort guidelines in Section 4.1 serve as a minimum standard, while the need for computational faithfulness tests or wet-lab validation depends on the task:**
> - **For tasks that are model-centered** (e.g., use IML for model sanity checking, debugging, or comparison), faithfulness checks in Section 4.2 are essential to ensure that the evidence from IML truly reflects the model's reasoning. For some studies in this category that demand stronger evidence, biological experiments (Section 4.3) may also be necessary. In such cases, lower-effort checks from 4.1 and 4.2 should act as gatekeepers to minimize wasted experimental resources.
> - **For tasks that are biology-centered** (e.g., use IML for biological discovery), experimental validation in 4.3 is the gold standard. Still, 4.1 and 4.2 remain necessary to filter out unreliable hypotheses before expensive experiments.
>
> To demonstrate why lower-effort actions remain necessary alongside higher-tier ones: Table 1 shows almost all IML methods fail to recover valid motifs for SP1 and TBP. Skipping the low-effort agreement (4.1) and faithfulness (4.2) checks leaves practitioners without warning of spurious hypotheses. However, running these checks reveals remarkably low scores, immediately flagging high-risk outputs and preventing wasted experimental resources.
>
> **Table 1:** Model performance and inter-method agreement, faithfulness (insertion-AUC), as well as biological alignment averaged across 4 different IML methods on TF binding prediction.
> TF|NTv3 Performance (F1↗)|IML Agreement(↗)|IML Faithfulness(↗)|IML Biological Alignment(↗)
> -|-|-|-|-
> CTCF|0.95|0.49(±0.19)|0.89(±0.07)|0.62(±0.21)
> MAX|0.88|0.35(±0.16)|0.87(±0.10)|0.28(±0.16)
> SP1|0.87|**0.15(±0.18)**|**0.26(±0.19)**|**0.09(±0.14)**
> TBP|0.85|**0.20(±0.18)**|**0.59(±0.25)**|**0.17(±0.13)**
> GATA1|0.94|0.20(±0.19)|0.42(±0.31)|0.27(±0.11)
>
> **`Action 3:`** We will revise Section 4 to include the above discussions to facilitate the practical adoption of our recommendations.

---

> > ### Author Rebuttal · Reviewer_vbqK · 2026-04-02
> >
> > The authors have addressed most of my concerns. Hence, I decide to keep positive.

---

### Official Review · Reviewer_6Mcn · 2026-03-14

**Significance:** 2
**Argument Clarity:** 2
**Rating:** 4
**Confidence:** 3

**Questions:**

It would be helpful to understand whether the observed interpretability failures generalize to other genomics tasks beyond transcription factor binding prediction. Another question is whether certain model architectures or training strategies produce more reliable explanations than others. Finally, since biological mechanisms can be complex and involve interactions beyond single motifs, it would be interesting to explore whether alternative definitions of biological ground truth could lead to different conclusions about explanation quality.

**Alternative Views Section:**

Yes

**Compliance With Llm Reviewing Policy A Conservative:**

Affirmed.

**Discussion Potential:**

2

**Paper Summary:**

The paper examines the reliability of interpretable machine learning (IML) methods used to explain deep learning models in genomics. Through a large-scale mapping study of 3,575 papers and a benchmarking experiment on transcription factor binding prediction, the authors show that current practices often rely on a single interpretation method, subjective validation, and cherry-picked successful examples. Their experiments demonstrate that different IML methods frequently produce inconsistent explanations, often fail to reflect the model’s true decision process, and may not align with known biological mechanisms. To address these issues, the authors propose a tiered evaluation framework that includes minimal reporting standards, computational validation, and experimental validation to ensure more rigorous and reproducible interpretability studies in genomics.

**Position:**

Yes

**Position In Title:**

Yes

**Related Work:**

3

**Strengths And Weaknesses:**

- Strengths

The paper provides a strong empirical critique of current interpretability practices in genomics by combining a large-scale literature analysis with systematic benchmarking experiments. It clearly demonstrates several important failure modes of popular attribution methods, including inconsistency across methods and lack of biological alignment. In addition, the proposed tiered evaluation framework offers practical and actionable guidelines that researchers can adopt to improve the rigor and reliability of interpretability studies

- Weaknesses

The study mainly focuses on feature attribution methods and a single application domain—transcription factor binding prediction—which may limit the generality of its conclusions for other interpretability approaches or genomic tasks. Furthermore, while the proposed evaluation guidelines are useful conceptually, the paper does not provide standardized benchmarks or tools that would make them easier to implement in practice. Additional experiments across more datasets and biological tasks could further strengthen the claims.

**Support:**

2

---

> ### Author Rebuttal · Authors · 2026-03-27
>
> ***Abbreviations:** **"W"** for Weaknesses, **"Q"** for Questions.*
>
> We thank the reviewer for the constructive feedback. Below we address each comment in detail.
>
> ---
> ### **W1/Q1: Generalization Beyond Feature Attribution**
>
> We thank the reviewer for this constructive suggestion. To test whether our findings generalize beyond feature attribution, we applied the Section 3 evaluation workflow to [a tRNA structural contact prediction task used in the nucleotide dependency analysis (NDA) paper](https://www.nature.com/articles/s41588-025-02347-3), which requires IML methods to identify pairwise interaction effects. We evaluated 3 IML methods: NDA, categorical Jacobians and [SQUID](https://www.nature.com/articles/s42256-024-00851-5).
>
> As shown in Table 1, **our previous findings (e.g., disagreement, faithfulness, and biological alignment issues) persist on this interaction task with new IML methods**.
>
> **Table 1:** Inter-method agreement, faithfulness (insertion-AUC) and biological alignment for 3 IML methods on tRNA structural contact prediction.
> |IML|IML Agreement(↗)|Faithfulness(↗)|Biological Alignment(↗)
> |---|---|---|---
> |NDA|0.25(±0.18)|0.91(±0.31)|0.62(±0.39)
> |Categorical Jacobians|0.14(±0.09)|0.72(±0.42)|0.26(±0.11)
> |SQUID|0.23(±0.17)|0.73(±0.39)|0.35(±0.16)
>
> ---
> ### **W2/Q1: Generalization Beyond TF Binding Prediction**
>
> We appreciate the reviewer's thoughtful advice. We have now extended our evaluation to splice site prediction. Similar to Table 1, the results in Table 2 **further confirm that our findings are generalizable beyond TF binding prediction. Thus, the central concern raised in this position paper is not specific to one task or one set of IML methods.**
>
> **Table 2:** Model performance and inter-method agreement, faithfulness (insertion-AUC), and biological alignment averaged across 4 different IML methods on splice site prediction.
>
> |Model|Precision(↗)|IML Agreement(↗)|IML Faithfulness(↗)|IML Biological Alignment(↗)
> |---|---|---|---|---
> |SpliceAI-10k|0.85|0.18(±0.09)|0.77(±0.11)|0.19(±0.08)
> |SpTransformer|0.84|0.23(±0.12)|0.84(±0.16)|0.27(±0.11)
> |SpliceBERT|0.85|0.20(±0.07)|0.85(±0.12)|0.28(±0.09)
>
> ---
> ### **W3: Standardized Benchmarks or Tools**
> We thank the reviewer for this suggestion. We have released an anonymous code and data repository at [this link](https://anonymous.4open.science/r/XAI4Gen-5D6C/), including all datasets and source code for IMLs and evaluations.
>
> Additionally, we are integrating these artifacts into a unified Python library to facilitate broader adoption.
>
> ---
> ### **Q2: Impact of Model Architectures or Training Strategies**
>
> We thank the reviewer for this point. Our experiments (Section 3, Table 2) evaluate diverse architectures (including Transformers, Hyena, CNNs, and hybrids) and suggest that **no specific model or training strategy consistently yields more reliable explanations.** Instead, the identified failure modes persist across settings.
>
> These findings reinforce our claim that rigorous evaluation is essential for IML, regardless of architecture or training strategy. While we agree that isolating these factors via controlled ablations is a valuable future direction, it is non-trivial due to **confounding factors (e.g., pretraining data)** and lies beyond the scope of this position paper.
>
> That said, prior work suggests targeted training can improve explanation reliability. For instance, incorporating axiomatic attribution priors during optimization [has been shown to](https://www.nature.com/articles/s42256-021-00343-w) improves faithfulness. We will discuss these approaches in the revised manuscript to better contextualize how design choices influence interpretability.
>
> ---
> ### **Q3: Alternative Definitions for Biological Ground Truth**
>
> The reviewer raises an interesting point. To investigate this, we expanded our ground truth beyond single UniBind peaks. Using PWMs from the JASPAR database, we scanned test sequences for "multi-hit" occurrences by redefining the ground truth as all statistically enriched primary and co-factor motifs within the window.
>
> Table 3 shows this expanded regulatory grammar improves alignment for CTCF, MAX, and GATA1. This confirms the hypothesis that IML methods can capture broader, valid biological context rather than a single motif. However, SP1 and TBP performance remains near zero. Thus, while broader definitions reveal successes in high-signal cases, the fundamental vulnerability to compositional bias persists. **This observation again highlights that although specific IML behaviors fluctuate depending on the experimental setup, the underlying failure modes we identified remain pervasive in practice.**
>
> **Table 3:** Biological alignment score averaged across 4 IML methods on NTv3 under different ground truth criteria.
> |Ground truth|CTCF|MAX|SP1|TBP|GATA1
> |---|---|---|---|---|---
> |UniBind|0.62(±0.21)|0.20(±0.16)|0.10(±0.14)|0.18(±0.13)|0.27(±0.11)
> |Multi-hit JASPAR|0.88(±0.28)|0.65(±0.43)|0.21(±0.24)|0.06(±0.22)|0.55(±0.47)

---

> > ### Author Rebuttal · Reviewer_6Mcn · 2026-04-03
> >
> > The authors addressed my concerns. Therefore, I decided to keep my score positive.

---

### Official Review · Reviewer_jDmk · 2026-03-16

**Significance:** 3
**Argument Clarity:** 2
**Rating:** 4
**Confidence:** 4

**Questions:**

Is it possible that the IML methods are just struggling because the current models are just not good enough? They find a hint of this in comparing transcription factors with strong motifs versus TFs with weaker motifs

Given that most methods fail on certain TFs, how should a practitioner interpret this failure across multiple IML methods?

**Alternative Views Section:**

Yes

**Compliance With Llm Reviewing Policy A Conservative:**

Affirmed.

**Discussion Potential:**

3

**Final Justification:**

Thanks to the authors for their clarification my overall impression is consistent and I think their analysis of demonstrating explainability method disagreement is useful to the community.

**Paper Summary:**

The authors examine use of IML in genomics from a few different perspectives. First, they examine how these methods are typically used in existing publications, finding that scientists usually use interpretability methods for identification of positive instances, matching up with the model's conclusion about known biology. Typically, in publications, just a single IML method is used. In Section 3, they conduct an evaluation of IML methods on a few different models for a number of transcription factor binding motifs. They find low concordance between different IML methods. They then conduct faithfulness tests, where they either remove or add nucleotides that correspond to known true underlying motifs. They find that, between the four IML methods evaluated, they are not often able to reproduce the biological illogrand truth. The paper concludes with practical guidelines recommending multiple use of IML methods including taking the intersection between performant methods.

**Position:**

Yes

**Position In Title:**

Yes

**Related Work:**

2

**Strengths And Weaknesses:**

### Strengths
- The authors approach this IML evaluation from multiple angles, first examining ground truth reproduction, then examining consistency and performing faithfulness tests.
- The authors perform curation of how IML methods are actually used in literature from a global perspective.
- The authors provide a set of recommendations for how practitioners should engage with interpretability tests, and the differences are grounded in the global analyses performed in the earlier section.

### Weaknesses
- The authors don't really mention that the methods are typically used in practice as complementary to some sort of performance metrics that are reported earlier in papers. This might explain some of the recommendations not being a perfect fit for how the IML methods are actually used in practice.
- I find some of the sections a little bit inconsistent with one another. For example, the authors find that IML tests are unable to reproduce known positive ground truth and yet recommend performing multiple IML tests and reporting the agreement. If certain IML tests are unable to reproduce the ground truth, then the recommendation of using overlapping tests will result in further degraded performance.
- The authors omit discussion on improving ICML methods. Techniques such as categorical Jacobians are connected to in silico mutagenesis and are used more prevalently in genomic language models. The authors don't also mention techniques, newer techniques from LLM literature, such as sparse autoencoders.

**Support:**

3

---

> ### Author Rebuttal · Authors · 2026-03-27
>
> ***Abbreviations:** **"W"** for Weaknesses, **"Q"** for Questions.*
>
> We thank the reviewer for their constructive feedback. Below, we address each comment in detail.
>
> ---
> ### **W1: Regarding Performance Metrics**
>
> We agree IML typically follows predictive performance, and poorly performing models often yield unreliable IML outputs (see DNABERT2 in Table 1 below). Our guidelines are designed to be deployed *after* model evaluation, and we will make this clear in the revision.
>
> However, as discussed in Alternative Views, well-performing models still require explanation-specific validation: **predictive success guarantees a working model, but not that post hoc explanations built upon it faithfully reflect its reasoning or true biological mechanisms** (e.g., see CTCF and GATA1 in Table 1).
>
> **Table 1.** Predictive performance of models across 3 TFs, alongside the biological alignment of IML methods. Setups are the same as in Section 3.
> Model|TF|F1 Score|DeepLIFT|IG|ISM|LIME
> -|-|-|-|-|-|-
> HyenaDNA|CTCF|0.92|0.43|0.33|0.68|0.56
> HyenaDNA|SP1|0.79|0.02|0.02|0.01|0.02
> HyenaDNA|GATA1|0.92|0.21|0.19|0.32|0.27
> DNABERT2|CTCF|0.82|0.02|0.02|0.56|0.53
> DNABERT2|SP1|0.77|0.00|0.00|0.05|0.11
> DNABERT2|GATA1|0.82|0.00|0.00|0.20|0.22
> NTv3|CTCF|0.95|0.54|0.71|0.66|0.59
> NTv3|SP1|0.88|0.03|0.25|0.05|0.07
> NTv3|GATA1|0.94|0.08|0.23|0.18|0.23
>
> ---
> ### **W2: Clarifying the Recommendation for Overlapping IML Methods**
>
> We apologize if our phrasing implied we recommend using IML overlaps for final decisions. As noted in G1 (L324–327), our actual intent is to **advocate for multiple IMLs as a *diagnostic tool* to measure disagreement for warning signal, not as an *ensembling tool* to improve performance.** Agreement only quantifies explanation instability, and even high agreement still requires faithfulness/biological validation.
>
> This is analogous to ML uncertainty quantification: low uncertainty doesn't guarantee good predictions, but high uncertainty is an immediate warning. For example, Table 2 shows biological alignment declines as IML method agreement decreases.
>
> **Table 2:** Mean biological alignment across 4 IML methods on the NTv3 model, stratified by IML agreement (tertiles: Low, Mid, High).
>
> |IML Agreement|CTCF|MAX|SP1|TBP|GATA1|
> |-|-|-|-|-|-|
> |Low|0.50(±0.15)|0.08(±0.12)|0.09(±0.05)|0.05(±0.05)|0.14(±0.08)|
> |Mid|0.64(±0.17)|0.18(±0.08)|0.11(±0.06)|0.05(±0.06)|0.18(±0.05)|
> |High|0.73(±0.12)|0.33(±0.06)|0.13(±0.04)|0.07(±0.03)|0.22(±0.04)|
>
> ---
> ### **W3: Newer IML Methods**
>
> We thank the reviewer for this suggestion. We amended experiments using categorical Jacobians, and **our Section 3 findings persist**.
>
> For example, Table 3 shows strong disagreement between categorical Jacobians and other IML methods as well as low faithfulness scores on SP1 and TBP. These results also reconfirmed our above observation that low inter-method agreement is a warning signal. We will discuss these results in the revision.
>
> Sparse autoencoders (SAEs) belong to the representation-level interpretability family rather than input attribution. Thus, they fall outside the apples-to-apples scope of our current audit. However, SAEs equally require rigorous validation, and we will highlight this in our future directions.
>
> **Table 3:** Additional results on TF binding prediction with categorical Jacobians and NTv3.
> |TF|F1(↗)|IMLs Correlation(↗)|MoRF-AUC(↘)|Insertion-AUC(↗)|Biological Alignment(↗)|
> |---|---|---|---|---|---|
> |CTCF|0.95|0.57(±0.22)|0.06(±0.07)|0.88(±0.09)|0.60(±0.17)|
> |MAX|0.89|0.30(±0.23)|0.27(±0.16)|0.89(±0.08)|0.18(±0.18)|
> |SP1|0.88|0.19(±0.34)|0.34(±0.10)|0.18(±0.12)|0.05(±0.02)|
> |TBP|0.85|0.17(±0.19)|0.57(±0.18)|0.52(±0.26)|0.09(±0.04)|
> |GATA1|0.94|0.14(±0.13)|0.31(±0.20)|0.38(±0.30)|0.28(±0.15)|
>
> ---
> ### **Q1: Impact of Model Quality**
>
> We agree model quality contributes to the weaker IML results on harder TFs. Yet, **model quality alone cannot account for all the results**. For example, even for CTCF, which is the easiest TF among the 5 with high model F1 scores, IML methods still (a) disagree with each other (Fig. 2b); (b) differ in faithfulness (Fig. 3); and (c) fall short of ceiling in biological alignment (Fig. 4a).
>
> This points to two independent conditions for IML success:
> 1. The model must learn the mechanism.
> 2. The IML method must faithfully surface it.
>
> We agree with the reviewer's point regarding the first condition, and our contribution shows condition 2 fails independently, and advocate explicit check for it.
>
> ---
> ### **Q2: When All IML Methods Failed**
>
> If multiple IML methods fail to identify the same features, practitioners should interpret this as a signal that (a) the model may not have learned the expected biology for this task, or (b) current IML methods are insufficient for this context. **In either case, the practitioner should NOT draw biological conclusions from the IML outputs.** This is precisely the diagnostic value of multi-method comparison: it prevents false confidence.

---

> > ### Author Rebuttal · Reviewer_jDmk · 2026-04-02
> >
> > Thank you to the authors for a comprehensive rebuttal and additional evaluations
> >
> > There is still a piece of the argument that's missing for me. As the authors show in their experimental section all IML methods are imperfect and disagree. And yet the recommendation is treating high disagreement that the explanations provide as a warning signal that the explanations are unreliable. Isn't this a bit tautological? Should the recommendation be to avoid using certain IML methods?

---

### Decision · Program_Chairs · 2026-04-30

**Decision:**

Accept (spotlight)

**Comment:**

All reviews were positive about the paper. Everyone is in clear agreement that the work characterizes the state of interpretable ML in genomics well, and suggests appropriate remedies for field-wide issues.

The frank discussion of shortcomings is likely to be especially helpful in a cross-disciplinary field such as genomics, where people are unaware of the limitations of interpretablity or what best practices should be. It is also of interest to the ICML community.